# Auxin efflux by PIN-FORMED proteins is activated by two different protein kinases, D6 PROTEIN KINASE and PINOID

Melina Zourelidou[1†], Birgit Absmanner[2†], Benjamin Weller[1], Inês CR Barbosa[1], Björn C Willige[1‡], Astrid Fastner[2], Verena Streit[1], Sarah A Port[1§], Jean Colcombet[3], Sergio de la Fuente van Bentem[3,4¶], Heribert Hirt[3#], Bernhard Kuster[5], Waltraud X Schulze[6,7], Ulrich Z Hammes[2*], Claus Schwechheimer[1*]

[1]Department of Plant Systems Biology, Technische Universität München, Freising, Germany; [2]Department of Cell Biology and Plant Biochemistry, Universität Regensburg, Regensburg, Germany; [3]Unité de Recherche en Génomique Végétale, Université Evry, Evry, France; [4]Department of Plant Molecular Biology, Max F. Perutz Laboratories, University of Vienna, Vienna, Austria; [5]Proteomics and Bioanalytics, Technische Universität München, Freising, Germany; [6]Max-Planck-Institute of Molecular Plant Physiology, Potsdam, Germany; [7]Plant Systems Biology, Universität Hohenheim, Hohenheim, Germany

*For correspondence: ulrich.hammes@biologie.uni-regensburg.de (UZH); claus.schwechheimer@wzw.tum.de (CS)

†These authors contributed equally to this work

Present address: ‡Salk Institute for Biological Studies, La Jolla, United States; §Department of Molecular Biology, Göttingen University Medical Center, Göttingen, Germany; ¶Syngenta Seeds B.V, Enkhuizen, Netherlands; #Center for Desert Agriculture, King Abdullah University of Science and Technology, Thuwal, Saudi Arabia

Competing interests: The authors declare that no competing interests exist.

**Abstract** The development and morphology of vascular plants is critically determined by synthesis and proper distribution of the phytohormone auxin. The directed cell-to-cell distribution of auxin is achieved through a system of auxin influx and efflux transporters. PIN-FORMED (PIN) proteins are proposed auxin efflux transporters, and auxin fluxes can seemingly be predicted based on the—in many cells—asymmetric plasma membrane distribution of PINs. Here, we show in a heterologous *Xenopus* oocyte system as well as in *Arabidopsis thaliana* inflorescence stems that PIN-mediated auxin transport is directly activated by D6 PROTEIN KINASE (D6PK) and PINOID (PID)/WAG kinases of the Arabidopsis AGCVIII kinase family. At the same time, we reveal that D6PKs and PID have differential phosphosite preferences. Our study suggests that PIN activation by protein kinases is a crucial component of auxin transport control that must be taken into account to understand auxin distribution within the plant.

## Introduction

The synthesis and proper distribution of the hormone auxin within the growing plant body is essential for basically all differentiation processes throughout plant development as well as for the plant's tropic responses. As such, proper plant development and morphology strictly require the directed cell-to-cell transport of auxin, which is achieved by a system of auxin influx and efflux transporters (*Teale et al., 2006*). AUXIN RESISTANT1 (AUX1)/LIKE-AUX1 (LAX) proteins are auxin influx transporters and PIN-FORMED (PIN) proteins, which have been proposed to act in concert with ABC transporters, are the proposed auxin efflux transporters (*Galweiler et al., 1998*; *Friml et al., 2002*; *Noh et al., 2003*; *Geisler et al., 2005*; *Bainbridge et al., 2008*; *Peret et al., 2012*). The directed transport of auxin throughout the plant is critically determined by the—in many cells–asymmetric plasma membrane distribution of PINs and plant developmental processes have been successfully modeled based on the knowledge of PIN distribution and PIN protein behavior (*Jonsson et al., 2006*; *Smith et al., 2006*; *Wisniewska et al., 2006*; *Blakeslee et al., 2007*; *Grieneisen et al., 2007*).

**eLife digest** In plants, a hormone called auxin controls the growth of the stems and roots. This chemical is transported from cell to cell, and its flow though the plant is redirected continuously as the plant is developing. Auxin is pumped out of cells by proteins in the cell membrane called 'auxin efflux carriers'. These proteins are usually found on one side of each cell and this is what gives the direction to auxin transport.

Zourelidou, Absmanner et al. now report that being positioned on the correct side of a plant cell is not enough to enable an efflux carrier to do its job—it must also be turned on by kinases before it can pump auxin out of cells. Kinases are enzymes that add phosphate groups to specific sites on other proteins, and plants without certain kinases are unable to transport auxin.

When Zourelidou, Absmanner et al. produced the efflux carrier and a plant kinase—which turns the efflux carrier on—in immature egg cells from frogs, auxin was rapidly pumped out of the cells. However, cells that contained the efflux carrier but not the kinase could not transport the hormone. Importantly egg cells from frogs do not normally transport auxin, but these cells are commonly used in experiments because they are large, which makes them easier to work with in the lab.

One of at least two kinases must tag a number of sites on the efflux carrier to ensure that it is switched on. It was already known that some of these sites are involved in making sure that the efflux carrier is located on the correct side of the cell. Zourelidou, Absmanner et al. also found that auxin itself encourages the addition of phosphate groups onto the efflux carrier.

Though it was thought that knowing where the auxin transporters are was enough to explain the direction of auxin transport in plants, it is now clear that activation by the kinases needs to be taken into account too. And since these kinases may activate the transporters to different extents, identifying how these proteins are controlled, for example by auxin itself, will be the next challenge in the field.

We have previously identified and studied Arabidopsis protein kinases of the AGCVIII family designated D6 PROTEIN KINASE (D6PK) (*Zourelidou et al., 2009*). The D6PK family is comprised of four functionally redundant members, namely D6PK, D6PK-LIKE1 (D6PKL1), D6PKL2 and D6PKL3. Although D6PKs are devoid of any sequence features indicative for an association of these protein kinases with the plasma membrane, D6PKs colocalize with PIN proteins at the basal (rootward) plasma membrane in cells of the root cortex and stele, the hypocotyl and main inflorescence stem as well as the shoot apical meristem (*Zourelidou et al., 2009*; *Barbosa et al., 2014*). D6PKs phosphorylate PIN proteins in vitro and PIN phosphorylation is reduced in *d6pk* mutants in vivo without affecting PIN distribution or strongly affecting PIN abundance (*Zourelidou et al., 2009*; *Willige et al., 2013*; *Barbosa et al., 2014*). Just as the PINs, D6PK constitutively cycles intracellularly between endosomal compartments and the plasma membrane but both, PINs and D6PK, traffic via distinct intracellular routes and seemingly encounter each other only at the basal plasma membrane (*Barbosa et al., 2014*). Since PIN phosphorylation, as assessed by evaluating overall PIN1 and PIN3 phosphorylation levels, rapidly reacts to the presence and absence of D6PK at the plasma membrane, we postulated that D6PKs directly activate auxin transport by PIN phosphorylation (*Willige et al., 2013*; *Barbosa et al., 2014*). This hypothesis has, however, never been tested.

Another subfamily of AGCVIII kinases comprises the proteins PINOID (PID), WAG1, and WAG2 (*Christensen et al., 2000*; *Benjamins et al., 2001*; *Santner and Watson, 2006*; *Galvan-Ampudia and Offringa, 2007*). Phosphorylation of PINs by PID/WAGs has previously been proposed to control PIN polarity (*Friml et al., 2004*; *Michniewicz et al., 2007*; *Dhonukshe et al., 2010*; *Huang et al., 2010*). PID/WAGs phosphorylate PINs at three highly conserved phosphosites, designated S1–S3 (*Dhonukshe et al., 2010*; *Huang et al., 2010*). Modulating PIN phosphorylation either by PID or WAG overexpression or by introducing phosphorylation-mimicking mutants in PIN1 seemingly results in a basal-to-apical shift in PIN polar distribution (*Michniewicz et al., 2007*; *Dhonukshe et al., 2010*; *Huang et al., 2010*). The proposed loss of PIN phosphorylation in the *pid* mutant has been used to explain the phenotypic similarity between *pin1* and *pid* mutants: *pin1* mutants, on the one side, have a pin-formed inflorescence because they are devoid of the central auxin efflux protein required for shoot meristem differentiation (*Galweiler et al., 1998*); *pid* mutants, on the other side, are deficient in PIN1 phosphorylation, which seemingly prevents the essential basal-to-apical polarity switch required to redirect auxin fluxes during differentiation at the shoot meristem (*Friml et al., 2004*).

The PID/WAG-mediated repolarization of PIN proteins is also important for phototropic responses (*Ding et al., 2011*). During phototropic bending of the hypocotyl, the polarity of the relevant PIN3 protein changes upon light exposure and this polarity switch is required for auxin redistribution in the hypocotyl and for efficient phototropism. This PIN3 polarity change requires the activity of PID/WAG protein kinases and it has been proposed that PID/WAG-dependent PIN3 phosphorylations directly control this process (*Ding et al., 2011*). We showed previously that D6PKs also play a critical role in this process: *d6pk* mutants are strongly impaired in phototropic hypocotyl bending and the inability of *d6pk* mutants to efficiently transport auxin from the cotyledons to the hypocotyl may be responsible for this tropism defect (*Willige et al., 2013*). Importantly, the light-induced and PID/WAG-dependent PIN3 polarity changes required for hypocotyl bending can still take place in the absence of *D6PKs* suggesting that the function of PID/WAGs in auxin transport and phototropism can be uncoupled from that of the D6PKs and that both kinases may control PINs independently and differentially (*Willige et al., 2013*). While the differential biological function of D6PK and PID/WAGs in the context of phototropism may be explained by the two kinases being active in different tissues or during different stages of the phototropism response, there is also evidence that the two kinases have differential biochemical activities. While the overexpression of PID and WAG kinases results in a basal-to-apical PIN shift, the overexpression of D6PKs does not affect PIN distribution (*Zourelidou et al., 2009*; *Dhonukshe et al., 2010*). Inversely, the loss of *PID* function results in strong differentiation defects of the primary inflorescence, which are not apparent in the *d6pk* mutants. Thus, there is evidence for a differential biochemical activity of D6PKs and PID/WAGs but the molecular basis of this differential activity remains to be determined.

The auxin efflux activity of PINs has previously been demonstrated by passive loading of yeast, plant, or mammalian cells with radiolabeled auxin (*Petrasek et al., 2006*; *Wisniewska et al., 2006*; *Mravec et al., 2008*; *Yang and Murphy, 2009*). In these experiments, the auxin efflux activity of PINs was deduced from the reduced amount of radiolabeled auxin that accumulated in cells (over-)expressing certain PIN proteins in comparison to control samples. Because these experiments used passive loading of auxin, it is unclear if the differences in intracellular auxin accumulation observed in these experiments are truly a result of differences in auxin efflux or a consequence of differences in auxin uptake. In other studies, auxin efflux was shown based on differences in auxin retention after passive loading and subsequent transfer to auxin-free medium, thereby reversing the electrochemical gradient. In these studies, background transport activities could not be ruled out and differences became apparent only at endpoint steady-state levels. To date, there has been no report of a heterologous expression system that allows measuring auxin export directly in the linear phase.

Here, we report the results from direct auxin efflux experiments with radiolabeled auxin (indole-3-acetic acid, IAA) injected into *Xenopus* oocytes. We find that PINs are unable to promote auxin efflux in this system unless PINs become activated by specific protein kinases of the Arabidopsis AGCVIII family. We map the phosphosites of these kinases in the PINs and further show that phosphorylation of conserved phosphosites is required for the efficient activation of PIN1 and PIN3. Our study strongly suggests that the activation of PIN-mediated auxin efflux by protein kinases is a crucial component of auxin transport control that must be taken into account to understand auxin distribution within the plant.

## Results

### D6PK is required for basipetal auxin transport in inflorescence stems

In *Arabidopsis thaliana*, the four AGCVIII kinases of the D6PK subfamily D6PK, D6PK-LIKE1 (D6PKL1), D6PKL2 and D6PKL3 redundantly control auxin transport-dependent growth (*Zourelidou et al., 2009*; *Willige et al., 2013*). Mutants with defects in multiple *D6PK* genes such as *d6pk d6pkl1* (*d6pk01*) double and *d6pk d6pkl1 d6pkl2* (*d6pk012*) triple mutants are severely impaired in several developmental processes including tropic responses (*d6pk01 and d6pk012*) and lateral root differentiation (*d6pk012*) (*Zourelidou et al., 2009*; *Willige et al., 2013*). In inflorescence stems, auxin is transported primarily in a basipetal (rootward) direction (*Teale et al., 2006*). To understand the contribution of the individual *D6PK* genes to auxin transport in inflorescence stems, we measured basipetal auxin transport in primary inflorescence stems of a selected set of *d6pk* single, double and triple mutants that represented a previously established phenotypic series (*Zourelidou et al., 2009*; *Willige et al., 2013*). In these experiments, we noted a decrease in auxin transport in mutants of increased

mutant complexity (**Figure 1**). While auxin transport defects were comparatively subtle in *d6pk* single mutants, the decrease in basipetal auxin transport was as strongly impaired in the *d6pk012* triple mutant as in mutants of *PIN1*, a major PIN protein in this tissue (**Figure 1**). Furthermore, we found that *D6PKs* are coexpressed with *PINs* in stems (**Figure 1—figure supplement 1**) and that both, D6PK and PIN1, localize to the basal plasma membrane in cells where auxin levels are high as suggested by the auxin response reporter *DR5:GFP* (**Figure 1—figure supplement 2**). Based on these observations, we concluded that D6PKs have an essential role in auxin transport regulation in inflorescence stems.

## D6PK activates PIN-mediated auxin efflux in a Xenopus oocyte system

Since auxin transport is impaired in *d6pk* mutant inflorescence stems and since we had previously accumulated evidence that D6PK directly phosphorylates PINs (**Zourelidou et al., 2009**; **Willige et al., 2013**; **Barbosa et al., 2014**), we hypothesized that D6PK may directly activate auxin transport by PIN phosphorylation in vivo. To test this hypothesis, we established a heterologous test system for measuring auxin efflux using Xenopus *laevis* oocytes. In this assay, in vitro transcribed cRNAs for the proteins under investigation were injected into the oocytes 4 days prior to the experiment to allow for protein synthesis. At the beginning of the experiment, radiolabeled IAA was injected and the amount of residual radiolabel was measured in the oocytes after incubation for up to 30 min. PIN as well as D6PK protein accumulated at the plasma membrane also in oocytes as shown by immunoblots for PINs and confocal microscopy for D6PK (**Figure 2A,B**). An inherent feature of this assay system was the gradual loss of the injected radiolabeled IAA from the oocytes over time–in the absence of exogenous proteins–which we attributed to the leakiness of the plasma membrane for IAA (**Figure 2C–F**). Interestingly, when we tested PIN1 or PIN3 alone, we did not observe any measurable auxin efflux that differed from the background, suggesting that the PINs are inactive auxin transporters in the oocyte system. However, when we co-expressed D6PK with the PINs we observed a significant and kinase

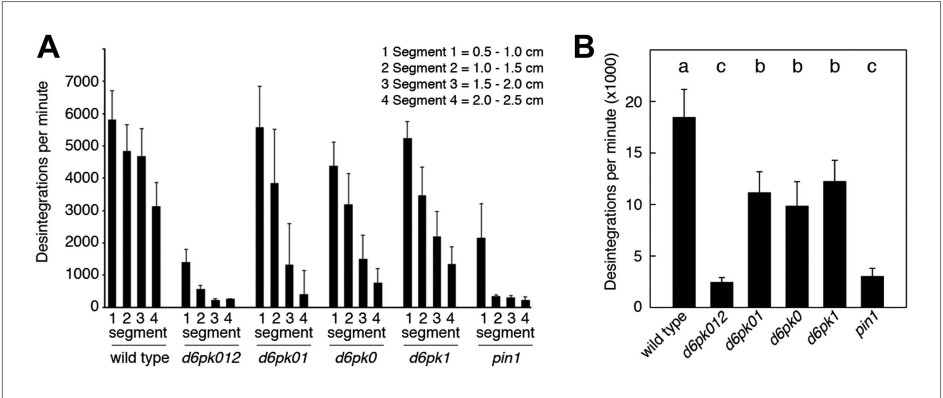

**Figure 1**. Basipetal auxin transport is impaired in *d6pk* and *pin1* mutants. (**A**) Basipetal auxin transport measured in inflorescence stems of 5-week-old Arabidopsis plants. Segment numbers refer to the 5 mm stem segments dissected from the primary inflorescence stem where segment 1 is the 5 mm segment closest to the radiolabeled auxin. The 5 mm segment directly in contact with the radiolabeled auxin is not included. Mutant nomenclature: *d6pk0, d6pk-1; d6pk1, d6pkl1-1; d6pk01, d6pk-1 d6pkl1; d6pk012, d6pk-1 d6pkl1 d6pkl2-2*. A linear mixed-effects model analysis (fixed factor) revealed statistically significant differences (p<0.01) in the transport rates between the wild type and all mutant genotypes, between the *d6pk* single mutants and the higher order *d6pk* mutants as well as between the *d6pk01* double mutant and the *d6pk012* triple mutant. *d6pk012* and *pin1* are not significantly different (p=0.43). (**B**) Amount of radiolabeled auxin found in all segments of the plants shown in (**A**). An ANOVA revealed highly significant differences between groups (p<0.001). An all-pairwise post hoc analysis (Holm-Sidak) allowed the assignment of three significance levels indicated by letters (p≤0.05 between levels).

The following figure supplements are available for figure 1:

**Figure supplement 1**. *D6PKs* and *PINs* are coexpressed in vascular bundles of inflorescence stems.

**Figure supplement 2**. D6PK and PIN1 localize to the basal plasma membrane in xylem parenchyma cells.

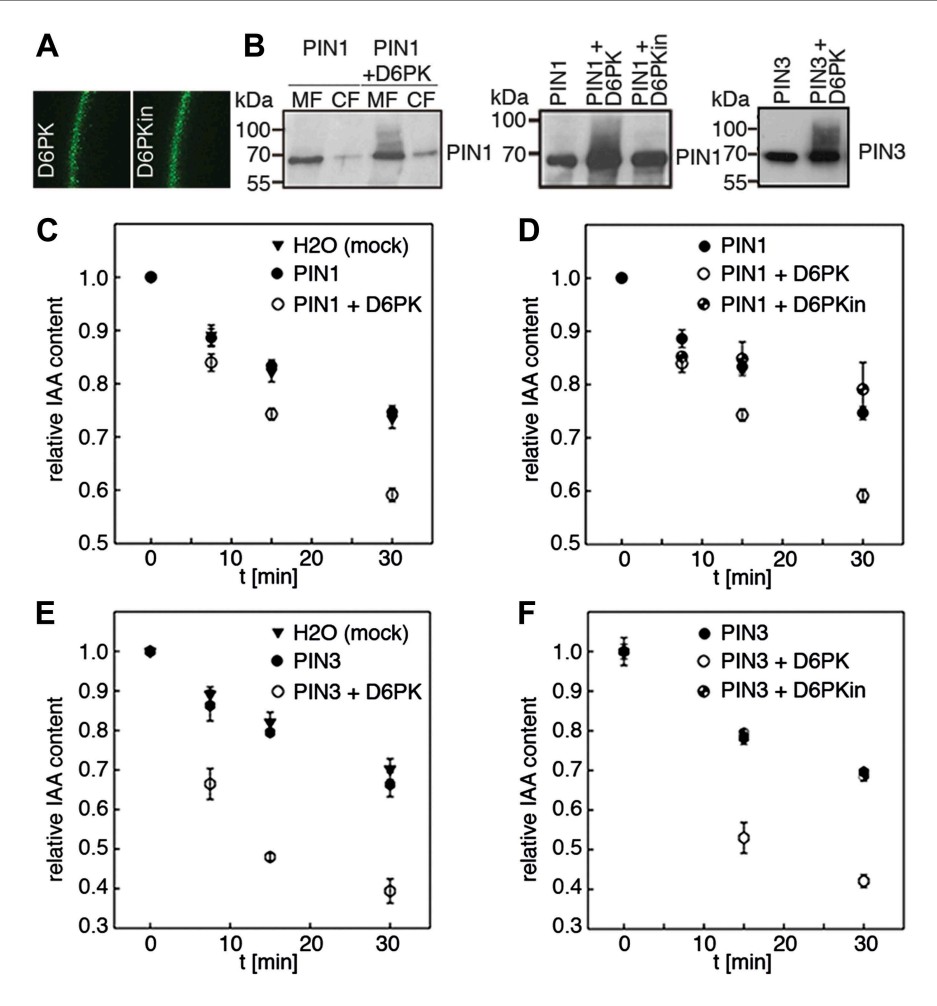

**Figure 2**. D6PK activates PIN-mediated auxin efflux in Xenopus oocytes. (**A**) Representative confocal microscopy images of oocytes expressing YFP:D6PK (D6PK) and YFP:D6PKin (D6PKin) reveals localization of the proteins at the plasma membrane. (**B**) Anti-PIN immunoblots of protein extracts from microsomal membrane (MF) fractions (and where applicable cytoplasmic fractions [CF]) from oocytes expressing PIN1, PIN3, YFP:D6PK (D6PK) and kinase-dead YFP:D6PK (D6PKin). (**C**)–(**F**). Results of representative auxin efflux assays conducted in Xenopus oocytes expressing PIN1, PIN3, YFP:D6PK (D6PK) and kinase-dead YFP:D6PKin as specified. Each data point represents the mean and standard error of at least 10 oocytes.

activity-dependent activation of auxin efflux. This activation correlated with the appearance of high molecular weight bands for PIN1 and PIN3 that appeared in anti-PIN immunoblots only in the presence of the active D6PK kinase (*Figure 2B*). In line with an activation of PINs by D6PK through direct PIN phosphorylation, a kinase-dead variant of D6PK could not activate auxin efflux in this system (*Figure 2D,F*). In summary, these experiments showed that D6PK is an activator of PIN-mediated auxin efflux in the oocyte expression system.

## D6PK phosphorylates PINs at specific phosphosites

Using mass spectrometry, we next identified D6PK-dependent phosphosites in the PINs after in vitro phosphorylation of the cytoplasmic loops (CL) of PIN1, PIN2, PIN3 and PIN4. These analyses resulted in the identification of two novel serine residues as conserved PIN phosphosites, S4 and S5, as well as three further serine phosphosites, S1–S3, that had previously been identified as phosphosites of the PID/WAG kinases (*Figure 3A*; *Figure 3—figure supplement 1*; *Figure 3—source data 1*; *Dhonukshe et al., 2010*; *Huang et al., 2010*). Whereas S1, S2 and S3 are conserved in all four PINs tested, S4 and

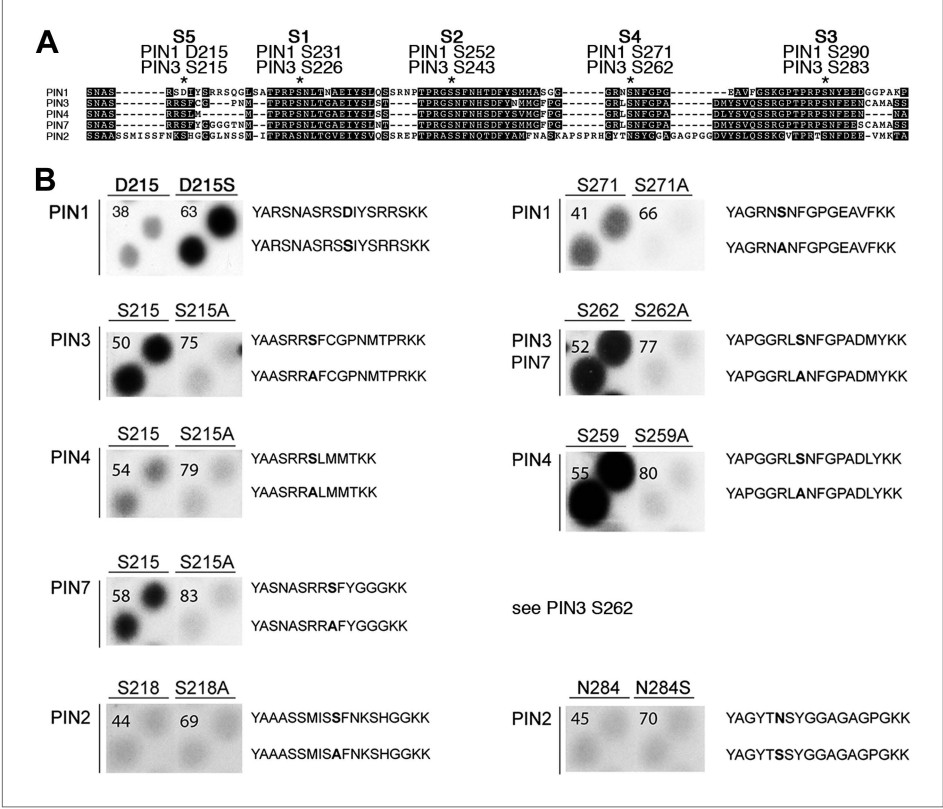

**Figure 3**. PIN S4 and S5 are phosphorylated by D6PK. (**A**) Sequence alignment of PIN cytoplasmic loop fragments indicating the PIN phosphosites identified after in vitro phosphorylation with D6PK. (**B**) Results of in vitro phosphorylation experiments with synthetic wild type and mutant peptides confirm the D6PK-dependent phosphorylation of sites corresponding to S4 (right panels) and S5 (left panels) in the PINs where the respective sites are conserved. PIN3 and PIN7 are sequence identical at the S4 phosphosite. Each reaction was spotted in duplicate. Amino acid sequences of the respective wild type and mutant peptides are shown on the right of each panel, their peptide identification numbers are shown in the upper left corner (*Supplementary file 1B*). The amino acid exchange in the respective peptide pair is shown in bold typeface. The N- and C-terminal amino acids Y–A and K–K were added to allow for peptide quantification after synthesis and to facilitate attachment of the peptide to the negatively charged P81 paper.

The following source data and figure supplements are available for figure 3:

**Source data 1**.

**Figure supplement 1**. Summary of the mass spectrometric analyses of PIN cytoplasmic loop phosphorylation by D6PK.

---

S5 are not conserved in PIN2 where the corresponding protein sequence motifs are divergent when compared to PIN1, PIN3, PIN4 and PIN7 and when compared to the strong conservation of the S1–S3 phosphosites in all PINs including PIN2 (*Figure 3A*). Furthermore, S5 was not conserved in PIN1 but aligned with a strongly conserved region of PIN1. At the position of S5, PIN1 had an aspartic acid (D; D215) and we speculated that D215 might be a natural phosphomimic variant of the S5 site (*Figure 3A*).

We next tested the identity and relevance of S4 and S5 in in vitro phosphorylation experiments using synthetic peptides as well as recombinant PIN CL fragments as substrates (*Figure 3B*). In the experiments with the synthetic peptides, we could confirm the identity and phosphorylation of the novel S4 and S5 phosphosites using mutant peptides as negative controls where the respective S had been replaced by an alanine (A) (*Figure 3B*). Since S5 from PIN3, PIN4 and PIN7 corresponded to D215 in PIN1 and since D215 was embedded in an otherwise highly conserved part of the protein, we were also interested in testing whether a serine (S) in a PIN1 D215S variant could be phosphorylated

by D6PK. Indeed, while a synthetic peptide comprising PIN1 D215 could not be phosphorylated by D6PK in vitro, the D215S peptide variant was efficiently phosphorylated indicating that, although the respective S5 phosphite was not conserved, the sequence conservation in this region was sufficient for phosphorylation by D6PK. This was suggestive for an overall structural conservation of this PIN1 protein domain (*Figure 3A*). In contrast, PIN2-specific peptides corresponding to the S4 or S5 phosphosites could not be phosphorylated by D6PK despite the fact that their sequences also contained serine residues. Phosphorylation of the corresponding peptides also failed when an asparagine (N) at the respective position was replaced by a serine (*Figure 3B*). Thus, S4 and S5 are novel PIN protein phosphosites that are differentially conserved in the five plasma membrane-resident PIN proteins with a role in promoting auxin efflux.

When we examined the contribution of the individual phosphosites to PIN1 phosphorylation in the context of the PIN1 cytoplasmic loop (CL) fragment, we found that PIN1 CL phosphorylation by D6PK was already strongly reduced (40% of wild type levels) in a mutant variant where only PIN1 S4 was replaced by an alanine (S4A; *Figure 4A*). In turn, mutations of the phosphosites PIN1 S1, PIN1 S2 or PIN1 S3 alone impaired phosphorylation by D6PK to a lesser extent (ca. 80%) and only mutation of all

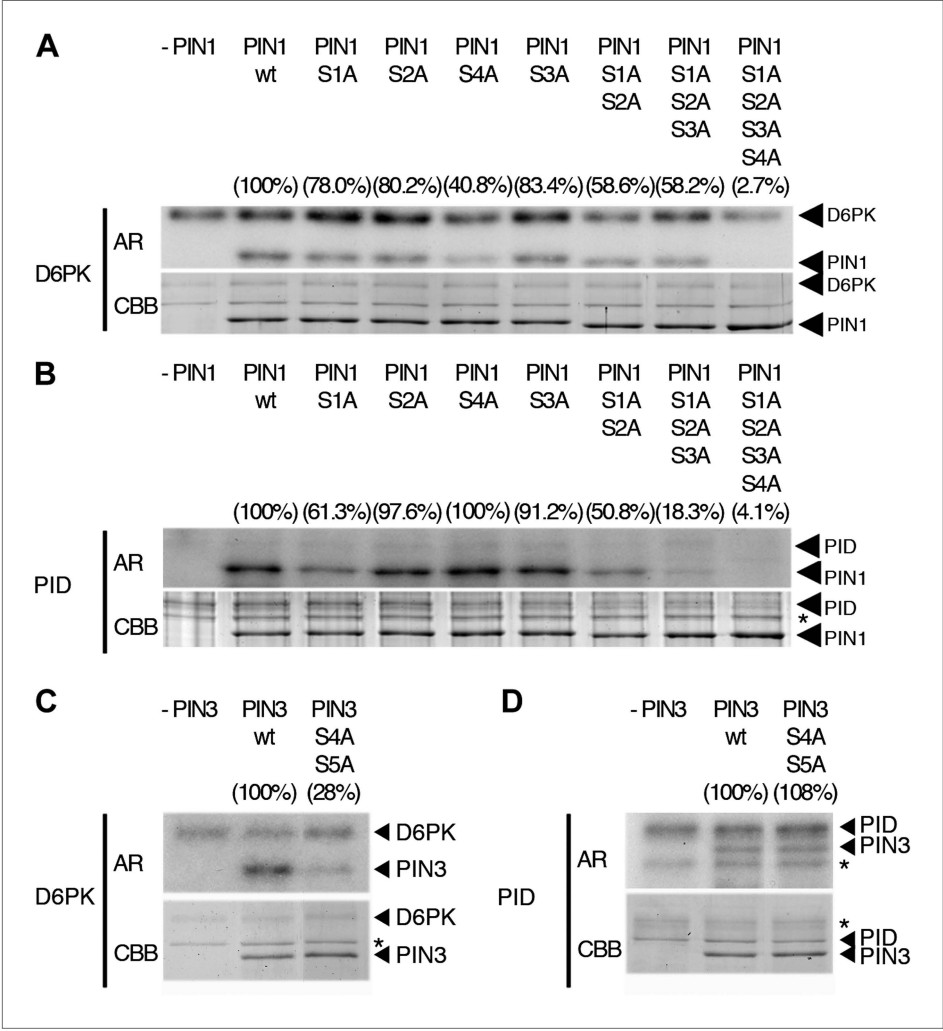

**Figure 4**. In vitro phosphorylation of PIN1 and PIN3. (**A**)–(**D**) Representative experiments with recombinant purified GST:D6PK (D6PK) or GST:PID (PID) and wild type or mutant PIN1 (**A** and **B**) or PIN3 (**C** and **D**) CL fragments in the presence of radiolabeled [©-$^{32}$P]ATP. AR, autoradiography; CBB, Coomassie Brilliant Blue-stained gel, loading control. Percentage values represent the amount of radiolabel incorporated into the PIN1 (**A**) and (**B**) and PIN3 (**C**) and (**D**) mutant proteins relative to the respective wild type protein after normalization to the loading control. Asterisks mark non-specific background bands or degradation products.

three sites in PIN1 S1A S2A S3A led to a clear reduction of PIN1 CL phosphorylation (58%; *Figure 4A*). Finally, the mutation of all four PIN1 phosphosites under investigation in PIN1 S1A S2A S3A S4A abolished phosphorylation by D6PK almost completely (2.7%; *Figure 4A*). Based on these analyses, we concluded that S4 is a major phosphosite for D6PK in PIN1.

Since PIN1 S1, S2, and S3 had previously been identified as phosphorylation targets of PID, we also examined and quantitatively compared the effects of the phosphosite mutations with those of D6PK. In the case of PID, the phosphorylation of PIN1 CL by PID was not altered in the PIN1 S4A mutant when compared to the wild type (100%; 40% for D6PK) but already strongly affected by the PIN1 S1A mutation (61%; ca. 80% for D6PK) and even more by PIN1 S1A S2A S3A (18%; 58% for D6PK; *Figure 4B*). Thus, D6PK and PID have an overlapping but also differential preference for specific phosphosites in PIN1. When we examined the effects of S4 and S5 site mutations in the context of PIN3, we detected a similar phosphosite preference. Whereas a PIN3 CL S4A S5A variant was still efficiently phosphorylated by PID its phosphorylation by D6PK was severely impaired (28%; *Figure 4C,D*). Thus, mutations of the five phosphosites have differential effects in the case of D6PK or PID.

## The S4 and S5 phosphosites are required for PIN activation by D6PK

We next evaluated the importance of S1–S3 and S4 for PIN1- and D6PK-dependent auxin efflux in oocytes. For this purpose, we calculated the transport rates of PIN1 and the S to A mutants as described in *Figure 5—figure supplement 1*. In line with the proposed important role of S4 for PIN1 phosphorylation, we found that a PIN1 S4A mutant was already significantly impaired in auxin efflux activation by D6PK in the auxin efflux experiments (*Figure 5A,B*). At the same time, the requirement of PIN1 S1, S2, and S3 for D6PK activation was not obvious with a PIN1 S1A S2A S3A mutant but became apparent in the presence of the S4A mutation where the PIN1 activation defect of the S4A mutation was further enhanced in the presence of mutations of the other three sites (*Figure 5A*). We thus concluded that PIN1 S4 is an important site for D6PK-dependent PIN1 activation but that all four phosphosites are required for full PIN1 activation. Also, in line with the results obtained in the in vitro phosphorylation experiments, we found that a PIN3 S4A S5A variant showed reduced responsiveness to D6PK when compared to wild type PIN3 providing further support for the importance of the S4 and S5 phosphosites for PIN activation by D6PK (*Figure 5C*).

Since S5 corresponded to an aspartic acid residue in PIN1 (D215) and because we could demonstrate that a peptide with a D215S replacement was efficiently phosphorylated by D6PK (*Figure 3*), we speculated that D215 might be a natural phosphomimic variant of the S5 phosphosite. We reasoned that PIN1 D215S might show a differential behavior in the auxin efflux experiments in the absence and presence of D6PK because the D215S mutant variant could show a stronger dependency on kinase activation. However, we found that the auxin efflux (activation) of the wild type PIN1 protein was indistinguishable from the behavior of the PIN1 D215S mutant in these oocyte experiments (*Figure 5D*). We therefore rejected this hypothesis.

To examine the biological significance of S4 and S5 for PIN function, we introduced wild type and mutant transgenes for the expression of *PIN1* and *PIN3* under the control of their respective promoters into *pin1* and *pin3 pin4 pin7* (*pin347*) mutants, respectively (*Figure 6*). In support of an important but not exclusive role of S4 phosphorylation for PIN1 function, we detected only a partial rescue of the auxin transport defect in inflorescence stems of *pin1* mutants transformed with PIN1 S4A compared to a full rescue with the wild type PIN1. While the mutant and the wild type transgene were able to complement the PID-dependent inflorescence differentiation defect of the *pin1* mutant (*Figure 6C,D*), D6PK-dependent basipetal auxin transport in the stem was compromised (*Figure 6A,B*). Since the mutation of the S4 phosphosite may potentially interfere with the polar distribution or the intracellular transport of the constantly trafficking PIN1 protein, we analyzed the polar distribution of PIN1 S4A and its sensitivity to the trafficking inhibitor Brefeldin A (BFA) (*Figure 6—figure supplement 1*). Since PIN1 S4A showed an identical behavior to wild type PIN1 in these experiments, we concluded that changes in PIN1 polarity, PIN1 trafficking or PIN1 abundance at the plasma membrane may not be causal for the observed differences in basipetal auxin transport. We also evaluated the effects of PIN3 phosphosite mutations using the ability of *PIN3* transgenes to complement the strong phototropism defect of the *pin3 pin4 pin7* (*pin347*) triple mutant (*Figure 6E,F*; *Willige et al., 2013*). When we measured the ability of wild type PIN3 and mutant PIN3 S4A S5A to complement the *pin347* mutant when expressed from a *PIN3* promoter fragment, we found that the phototropism defect of the *pin347* mutant was

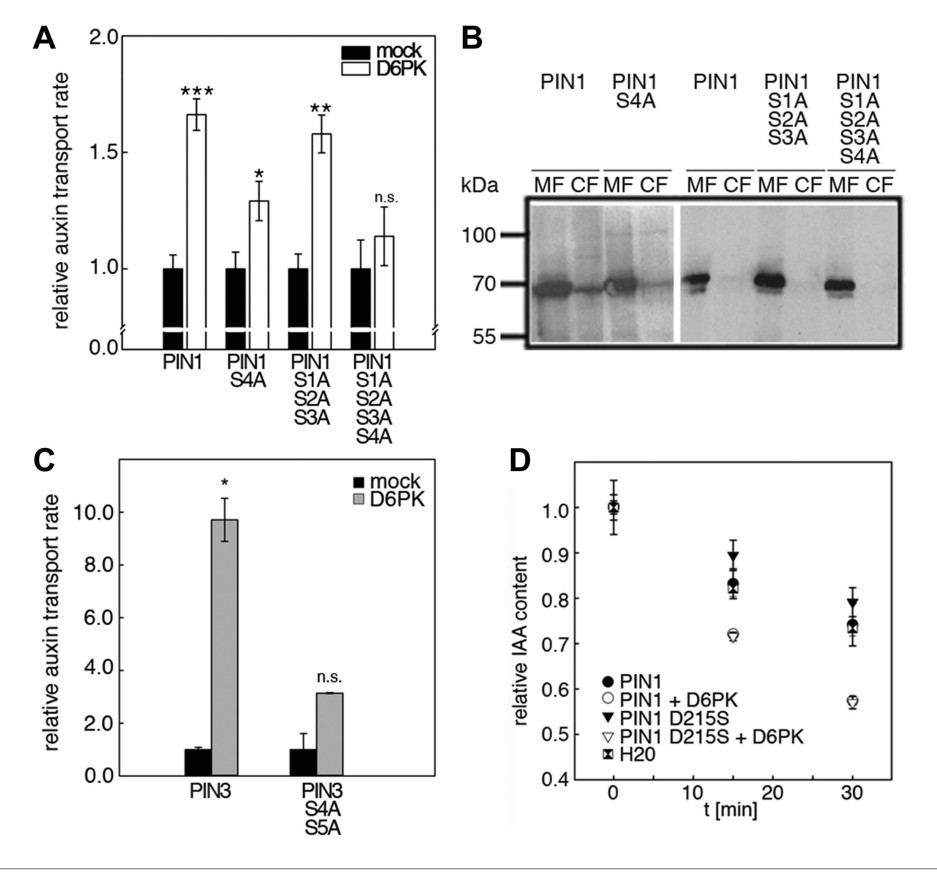

**Figure 5**. D6PK activates auxin transport through phosphorylation of specific serine residues. (**A**) Results of quantitative analyses from oocyte auxin efflux assays with D6PK and wild type or mutant PIN1. The averages of at least three independent measurements are shown after normalization to the mock control. Student's *t* test: \*p=0.022; \*\*p=0.005; \*\*\*p<0.001; n.s., not significant. (**B**) Anti-PIN1 immunoblots of microsomal membrane (MF) and cytoplasmic fractions (CF) of the corresponding oocytes used in (**A**). (**C**) PIN3 S4 S5 are required for full activation by D6PK. Results of quantitative analyses from oocyte auxin transport assays with D6PK and wild type PIN3 or the PIN3 S4A S5A mutant. The averages of at least three independent biological replicates are shown after normalization to the mock control. Student's *t* test \*, p=0.016; n.s., not significant. (**D**) PIN1 D215 does not contribute to the auxin transport activity of PIN1. Results of oocyte auxin efflux assays with wild type and mutant PIN1 together with YFP:D6PK (D6PK) as specified. Each data point represents the mean and standard error of measurements from at least 10 oocytes.

The following figure supplements are available for figure 5:

**Figure supplement 1**. Quantification of auxin efflux in Xenopus oocytes.

only partially complemented by the PIN3 S4A S5A transgene while it was fully complemented by wild type PIN3 (*Figure 6E*). This finding was in line with the hypothesis that D6PK-dependent PIN3 S4 and S5 phosphorylations are required for efficient basipetal auxin transport in the hypocotyls of dark-grown seedlings, which is a prerequisite for efficient hypocotyl bending. Consistent with the predominant role of the PID phosphosite phosphorylation at S1–S3, we found that the mutation of the PIN3 S1–S3 phosphosites as well as mutation of all five PIN3 phosphosites, S1–S5, fully impaired the ability of the PIN3 transgene to complement the *pin347* mutation (*Figure 6F*). This finding can be explained by the essential role of PID-dependent PIN3 polarity changes in the hypocotyl that take place after light exposure and that are required for phototropic hypocotyl bending. As we had previously shown, the PID-dependent PIN3 polarity change after phototropic stimulation is a distinct process that is independent from the regulation of basipetal auxin transport in the dark-grown seedling (*Ding et al., 2011*; *Willige et al., 2013*). In summary, this experiment supported the conclusion that the novel

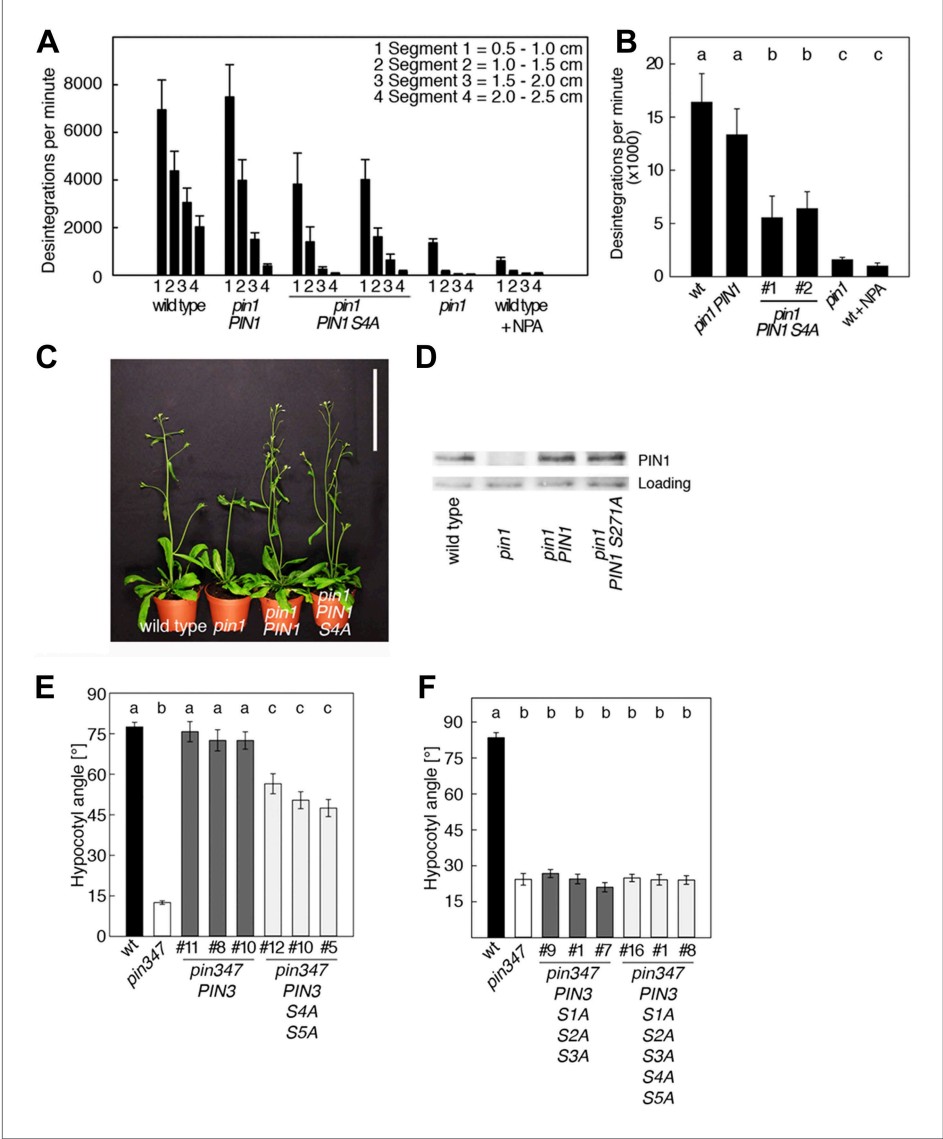

**Figure 6**. PIN1 S4 and PIN3 S4 S5 are required for full *pin* mutant complementation. (**A**) Basipetal auxin transport measured in inflorescence stems of 5-week-old Arabidopsis plants. Segment numbers refer to the 5 mm stem segments dissected from the inflorescence stem where segment 1 is the 5 mm segment closest to the radiolabeled auxin. The 5 mm segment directly in contact with the radiolabeled auxin was discarded. The values represent the mean and standard error of six biological replicates, except *pin1* and NPA-treated wild type (n = 2). A linear mixed-effects model analysis (fixed factor) revealed statistically significant differences (p<0.05) in the transport rates between the plant lines complemented with the PIN1 S4A construct and the other genotypes as indicated by the significance levels in (**B**). (**B**) Amount of radiolabeled auxin found in all segments of the plants shown in (**A**). An ANOVA revealed highly significant differences between groups (p<0.001). An all pairwise post hoc analysis (Holm-Sidak) allowed the assignment of three significance levels indicated by letters (p≤0.036 between levels). (**C**) Phenotypes of 5-week-old *pin1* mutants complemented with a transgenic construct expressing wild type *PIN1* and *PIN1 S4A* under control of a *PIN1* promoter fragment. Scale bar = 10 cm. (**D**) PIN1 immunoblot detects comparable PIN1 protein levels between the wild type and *PIN1* transgenic lines. (**E**) and (**F**) Analysis for the rescue of phototropic hypocotyl bending defects of a *pin3 pin4 pin7* mutant carrying wild type and mutant transgenes for the expression of wild type and mutant *PIN3* under control of a *PIN3* promoter fragment. Seedlings were exposed for 6 hr (**E**) or 20 hr (**F**) to unilateral white light before quantification. To assess differences between genotypes a Kruskal–Wallis ANOVA on ranks was performed. The differences in the median values among the different

*Figure 6. Continued on next page*

*Figure 6. Continued*

genotypic groups was highly significant (p<0.001). Different letters in indicate different significance levels (p<0.01) calculated by an all-pairwise multiple comparison (Dunn's Method).

The following figure supplements are available for figure 6:

**Figure supplement 1**. BFA-sensitivity of PIN1 and PIN1 S4A.

phosphosites, PIN3 S4 and S5, are required for full PIN3 activity, most likely by interfering primarily with basipetal auxin transport in the hypocotyls of dark-grown seedlings.

## S4 and S5 phosphorylation is strongly dependent on D6PK in vivo

Next, we were interested in examining the phosphorylation at PIN1 S4 and PIN3 S4 and S5 in vivo and to examine the phosphorylation at these sites in the presence and absence of D6PKs. To this end, we employed selected reaction monitoring (SRM), a mass spectrometry technique that allows detection and quantification of specific peptides and their phosphorylated variants in total protein preparations (*Picotti and Aebersold, 2012*). In these experiments, we detected a strong reduction in the in vivo abundance of the PIN1 S4 as well as PIN3 S4 phosphorylations that increased with increasing *d6pk* mutant complexity (*Figure 7A,B*). This decrease in S4 phosphorylation could not be explained by changes in the overall abundance of PIN proteins as shown by quantitative SRM analyses of the unphosphorylated PIN1 and PIN3 S4 peptides and analyses of internal control peptides (*Figure 7A,B*). Furthermore, introducing a *D6PK* transgene expressing *D6PK* under control of a *D6PK* promoter fragment rescued the PIN1 and PIN3 S4 phosphorylation defects (*Figure 7—figure supplements 1 and 2*). We also examined phosphorylation at PIN3 S5 using the same methodology and observed that the abundance of phosphorylation at these sites was as strongly reduced in the *d6pk012* triple mutant as observed for the S4 site. Again, the phosphorylation defect could not be explained by changes in PIN3 abundance and was rescued by a *D6PK* transgene as described above (*Figure 7—figure supplement 3*). Most importantly, the observed decreases in PIN1 and PIN3 phosphorylation were in good agreement with the reductions in auxin transport that we had detected in the same tissue of *d6pk* mutants (*Figure 1*). We therefore concluded that D6PKs are the major kinases targeting PIN1 S4, PIN3 S4, and PIN3 S5 in Arabidopsis inflorescence stems and that the reduced phosphorylation at these sites may be causal for the reduced auxin transport of *d6pk* mutants in this tissue.

## PID/WAG kinases also activate PINs

D6PKs belong to a larger family of AGCVIII kinases in Arabidopsis (*Galvan-Ampudia and Offringa, 2007*). Besides D6PKs and the already introduced PID/WAGs, other AGCVIII kinases such as the phototropin blue light receptors phot1 and phot2 as well as UNICORN (UCN) have known biological functions (*Inoue et al., 2008*; *Enugutti et al., 2012*). We were interested in testing the ability of these protein kinases to activate PIN-mediated auxin efflux and examined PID, WAG2 as well as phot1 and UCN together with PIN1 in the oocyte auxin transport assay (*Figure 8*). Interestingly, PID and WAG2 but not phot1 or UCN were able to activate PIN1-mediated auxin efflux (*Figure 8A,B*, *Figure 8—figure supplement 1*). We thus concluded that PID and WAG2 have a role in PIN activation besides their previously reported role in the control of PIN polarity (*Friml et al., 2004*; *Dhonukshe et al., 2010*).

We then examined whether the differential phosphosite preferences of D6PK and PID as observed in the in vitro phosphorylation experiments (*Figure 2C*) would also translate into differential defects in the oocyte auxin transport assay. Indeed, we found, in agreement with the in vitro data, that the PIN1 S1A S2A S3A mutant was less efficiently activated by PID than by D6PK (*Figure 8C*). Inversely, the PIN1 S4A mutation that strongly affected activation by D6PK did not significantly affect activation by PID. Again, mutation of all four PIN1 phosphosites, PIN1 S1A–S4A, resulted in the strongest impairment of PIN1 activation by PID (*Figure 8C*).

We also used SRM analyses to examine the effects of the loss of PID as well as WAG1 and WAG2 function on the phosphorylation of PIN1 S4 (*Figure 8—figure supplement 2*) and PIN3 S4 (*Figure 8—figure supplement 3*). However, in contrast to the strong defects in PIN S4 phosphorylation that we observed in the *d6pk* mutants, neither *pid* nor *wag1 wag2* mutants showed a clear reduction in PIN phosphorylation at the S4 phosphosite suggesting that PID and WAG1/WAG2 do not contribute to PIN S4 phosphorylation in this tissue. We also aimed to conduct the complementary SRM analysis

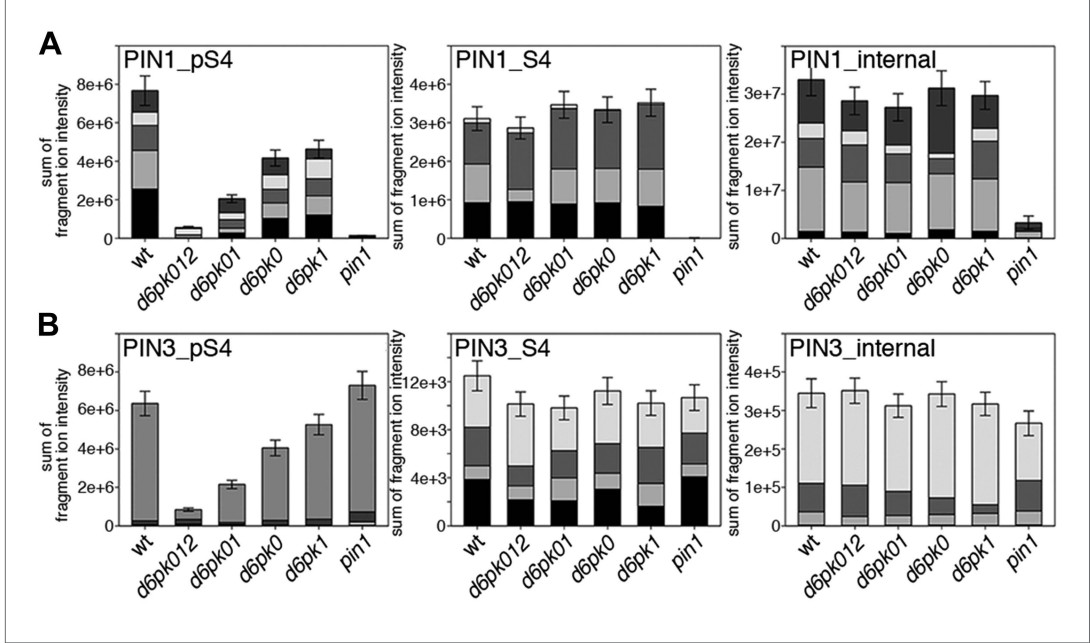

**Figure 7**. In vivo phosphorylation of PIN1 and PIN3. (**A**) and (**B**) Results of SRM analysis for the quantification of PIN1 S4 and PIN3 S4 phosphorylation in inflorescence tissue of 5-week-old Arabidopsis wild type and *d6pk* mutant plants (PIN1_pS4, phosphorylated form of PIN1 S4; PIN1_S4, unphosphorylated form etc). Internal peptides allow for an estimation of the overall PIN protein levels, standard deviations were calculated based on average variation of standard peptide abundances across all samples, columns represent the sum of individual fragment ions that are shown in different gray scales.

The following figure supplements are available for figure 7:

**Figure supplement 1**. Auxin-dependent phosphorylation at PIN1 S4.

**Figure supplement 2**. Auxin-dependent phosphorylation at PIN3 S4.

**Figure supplement 3**. Auxin-dependent phosphorylation at PIN3 S5.

experiment of the PIN1 and PIN3 S1, S2, and S3 phosphosites but, for technical reasons, had to restrict our efforts to SRM measurements of PIN1 S1 (*Figure 8—figure supplement 4*) and PIN3 S1 (*Figure 8—figure supplement 5*): Whereas the peptides comprising the S3 phosphosites of PIN1 and PIN3 were unsuitable for chemical peptide synthesis as predicted based on their primary amino acid sequence, we repeatedly failed to obtain synthetic peptides for the PIN1 and PIN3 S2 phosphosites. Our analysis of PIN1 and PIN3 S1 phosphorylations, however, showed that the phosphorylation at the S1 phosphosites was not affected when comparing the *d6pk012* mutant with the *d6pk012* mutant expressing a complementing *D6PK* transgene suggesting that D6PK does not contribute to the phosphorylation of S1 in vivo (*Figure 8—figure supplements 4 and 5*).

Since our phosphosite analyses indicated that D6PK and PID share their PIN target but display differential preferences for these phosphosites, we analyzed the functional redundancy of these two kinases in promoter swap experiments by expressing them under the control of the genes' promoter fragments in the *d6pk012* and the *pid* mutant background, respectively. These experiments demonstrated that D6PK and PID cannot functionally replace each other when expressed from the promoter of the respective other gene (*Figure 9*). Whereas the expression of *PID* from a *PID* promoter fragment was sufficient to complement the phenotypes of a *pid* mutant, the expression of *D6PK* under control of the *PID* promoter fragment failed to complement *pid* (*Figure 9A*). Inversely, *D6PK* but not *PID* expression from a *D6PK* promoter fragment was sufficient to complement the *d6pk012* mutant (*Figure 9B*). In summary, these genetic experiments supported our conclusion that D6PK and PID/WAGs are functionally divergent and these findings and conclusions are in line with previous

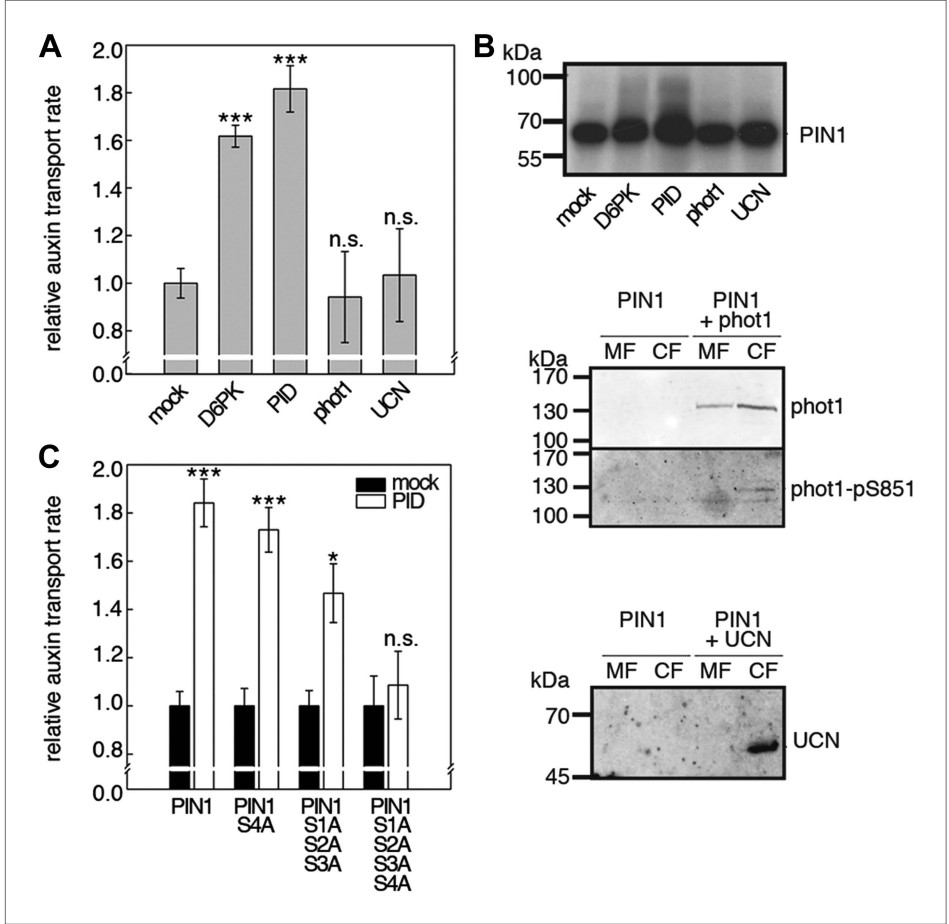

**Figure 8**. Capability of various AGCVIII kinases to active PIN1-mediated auxin efflux. (**A**) and (**C**) Results of quantitative auxin efflux assays performed in the oocyte system with PIN1 and different AGCVIII kinases (**A**) or mutant PIN1 and PID (**B**). The averages of at least three independent measurements, calculated as described in *Figure 5—figure supplement 1*, are shown after normalization to the mock control. In (**A**), a one-way ANOVA revealed high differences between groups (p<0.001) and a post hoc analysis (Holm-Sidak) indicated that the D6PK and PID values were significantly different from control oocytes (***p<0.001). In (**C**), a Student's t-test was performed: *p<0.027; ***p<0.001; n.s., not significant. (**B**) Immunoblots of total protein extracts prepared from oocytes expressing PIN1 and different AGC kinases. The presence and activation (phot1 only) of the non-effective kinases in the membrane (MF) and cytoplasmic fraction (CF) was confirmed with anti-phot1, anti-phot1-pS851 (for phot1 activation) and anti-UCN.

The following figure supplements are available for figure 8:

**Figure supplement 1**. WAG2 activates PIN1-mediated auxin transport.

**Figure supplement 2**. S4 phosphorylation in PIN1 is not strongly reduced in *pid* and *wag1 wag2* mutants.

**Figure supplement 3**. S4 phosphorylation in PIN3 is also not strongly reduced in *pid* and *wag1 wag2* mutants.

**Figure supplement 4**. Auxin-dependent phosphorylation at PIN1 S1.

**Figure supplement 5**. Auxin-dependent phosphorylation at PIN3 S1.

observations on the differential effects of these two kinases in PIN polarity control (*Dhonukshe et al., 2010*). These differential phosophosite preferences as detected in in vitro as well as in vivo phospho-site analyses may be the basis of the distinct roles of the two kinases in the control of PIN polarity and plant growth control.

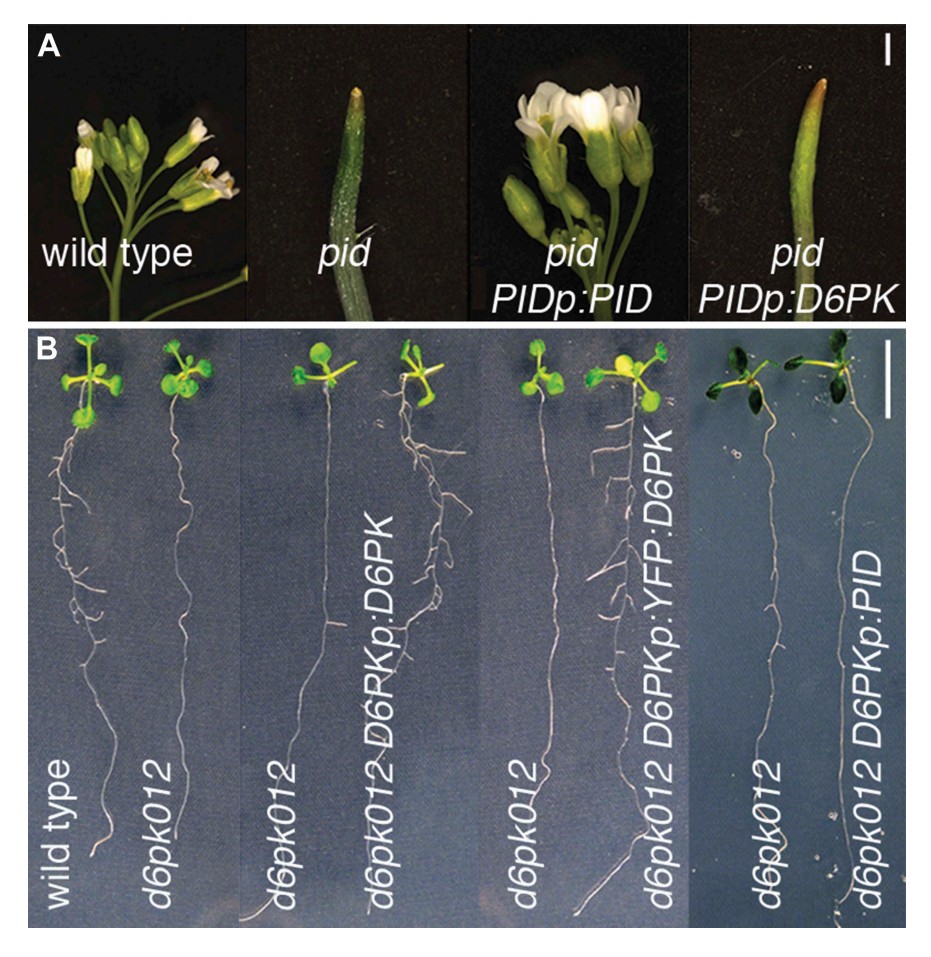

**Figure 9**. PID and D6PK are functionally non-redundant in vivo. (**A**) and (**B**) Test for genetic suppression of (**A**) the inflorescence phenotype of the *pid* mutant (5-week-old plants) and (**B**) the lateral root formation defect (8 day-old seedlings) of *d6pk012* triple mutants with *PID* and *D6PK* expressed from the *PID* (*PIDp*) and *D6PK* (*D6PKp*) promoters. The suppression of *d6pk012* by *D6PKp:YFP:D6PK* demonstrates the functionality of the YFP:D6PK translational fusion employed in other experiments. Scale bars = 1 mm (**A**) and 1 cm (**B**).

## Auxin promotes PIN phosphorylation

Since auxin had previously been shown to regulate auxin transport at the level of *PIN* transcription and PIN endocytosis control, we were also interested in examining the role of auxin on PIN phosphorylation. In these analyses, we detected concentration-, time- and D6PK-dependent increases in the phosphorylation of PIN1 S4, PIN3 S4 and PIN3 S5 already 15 min after auxin application (*Figure 10A,B*, *Figure 10—figure supplements 1–4*). While these increases were clearly observed in the wild type, only marginal increases in PIN phosphorylation at these sites were observed in the phosphorylation deficient *d6pk012* mutant. At the same time, phosphorylation at the preferential PID target site S1 was neither strongly impaired in *d6pk012* mutants when compared to a *d6pk012* mutant expressing a complementing *D6PK* transgene nor clearly induced by auxin (*Figure 8—figure supplements 4 and 5*). Furthermore, in agreement with an auxin-dependent control of PIN phosphorylation at S4 and S5, we detected increased phosphorylation at S4 and S5 in the auxin-overproducing *yucca* mutant (*Figure 10A,B*, *Figure 10—figure supplements 1–4*; *Zhao et al., 2001*). Although the analyses of the control peptides showed that there is also an overall increase in PIN abundance in *yucca*, the relative increases in phosphosite phosphorylations exceeded the increases in overall PIN abundance suggesting that PIN phosphorylation is activated in this mutant when compared to the wild type.

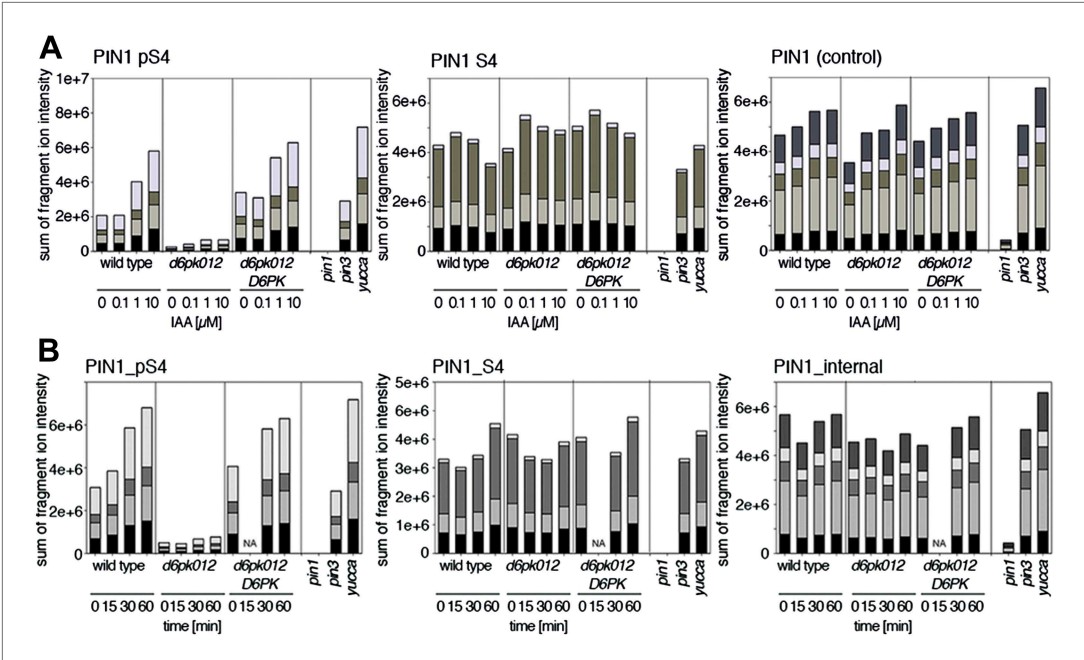

**Figure 10**. Dose- and time-dependent phosphorylation of PIN1 S4 after auxin treatment. (**A**). Quantification of PIN1 S4 phosphorylation in as a function of IAA concentration in inflorescence tissue of 5-week-old Arabidopsis plants. PIN1_pS4, phosphorylated form; PIN1_S4, unphosphorylated form. (**B**) SRM analysis of time-dependent S4 phosphorylation after IAA [10 µM] treatment of inflorescence tissue of 5-week-old Arabidopsis plants.

The following figure supplements are available for figure 10:

**Figure supplement 1**. Dose-dependent phosphorylation of PIN3 S4 after auxin treatment.

**Figure supplement 2**. Time-dependent phosphorylation of PIN3 S4 after auxin treatment.

**Figure supplement 3**. Dose-dependent phosphorylation of PIN3 S5 after auxin treatment.

**Figure supplement 4**. Time-dependent phosphorylation of PIN3 S5 after auxin treatment.

We had previously reported that D6PK is a plasma membrane-associated protein that cycles between the plasma membrane and the cytoplasm or intracellular compartments (*Zourelidou et al., 2009*; *Willige et al., 2013*; *Barbosa et al., 2014*). This cycling is highly sensitive to the trafficking inhibitor Brefeldin A (BFA) and in selected BFA-treatment conditions D6PK can be depleted from the plasma membrane without significantly affecting the plasma membrane abundance of PIN1 (*Figure 11*; *Barbosa et al., 2014*). The differential BFA-sensitivity of D6PK and PIN allowed us testing the contribution of plasma membrane-resident D6PK to PIN phosphorylation. For this purpose, we generated a phosphosite-specific antibody for PIN1 S4 that efficiently detected S4 phosphorylated PIN1 at the plasma membrane but failed to detect PIN1 S4A (*Figure 11—figure supplement 1*). Importantly, we found that PIN1 S4 phosphorylation was strongly decreased already minutes after BFA treatment when D6PK had become dissociated from the plasma membrane (*Figure 11*). Thus, PIN1 S4 phosphorylation depended on the presence of D6PK or other BFA-sensitive protein kinases at the plasma membrane.

## Discussion

In this study, we examined the functional roles of the D6PK protein kinases in PIN phosphorylation and auxin transport activation. We showed that *d6pk* mutants are impaired in basipetal auxin transport in inflorescence stems and postulated that PINs may be directly activated by D6PKs. This hypothesis was supported by the facts that D6PK colocalized with the basally localized PIN1 and PIN3 proteins in

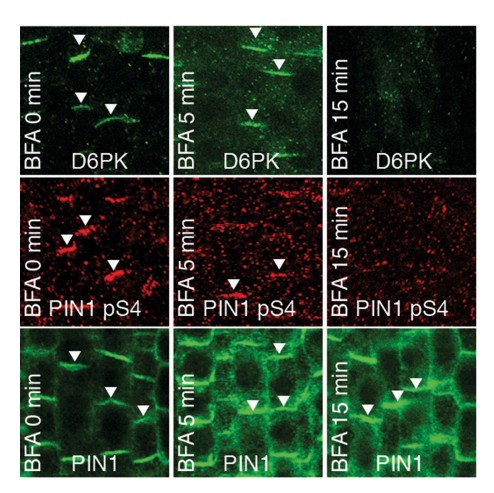

**Figure 11**. PIN1 pS4 is dependent on D6PK presence at the plasma membrane. Representative confocal images of root stele cells after immunostaining highlighting (arrowheads) the presence of YFP:D6PK (D6PK), S4-phosphorylated PIN1 (PIN1 pS4) and PIN1 at the plasma membrane before and the absence of D6PK and PIN1 pS4 after BFA treatment. Note that unphosphorylated PIN1 can still readily be detected in a polarized manner after S4-phosphorylation was efficiently removed.

The following figure supplements are available for figure 11:

**Figure supplement 1**. α-PIN1 pS4 is a PIN1 S4 phosphosite-specific antibody.

various cell types, that D6PKs phosphorylated PINs in vitro and that D6PKs influence PIN1 and PIN3 phosphorylation in vivo (*Zourelidou et al., 2009*; *Willige et al., 2013*; *Barbosa et al., 2014*). Here, we tested this hypothetical functional relationship by examining PIN1 or PIN3 activity and auxin transport at various levels. We showed that basipetal auxin transport was reduced in inflorescence stems of *d6pk* mutants and that PIN-mediated auxin efflux was activated by D6PK in *Xenopus* oocytes. Furthermore, we could rule out that the decreases in auxin transport as measured in inflorescence stems are the consequence of changes in PIN abundance as demonstrated using confocal imaging, immunoblotting, and SRM analyses of PIN proteins. We furthermore demonstrated that D6PK-dependent PIN activation was dependent on specific phosphosites in PIN1 and PIN3. Taken together, all our findings support the conclusion that D6PKs are major regulators of PIN-mediated auxin transport in inflorescence stems. Since *d6pk* mutants have a number of phenotypes such as gravitropism defects in the root, negative gravitropism defects in the hypocotyl, phototropism defects in the hypocotyl as well as defects in lateral root initiation (*Zourelidou et al., 2009*; *Willige et al., 2013*; *Barbosa et al., 2014*), we are tempted to speculate that these other *d6pk* mutant phenotypes are also the consequence of reduced auxin transport activity of PINs in the absence of the PIN-activating D6PK kinases.

The detailed analysis of the S1–S5 phosphosites in in vitro phosphorylation experiments and in oocyte auxin transport experiments revealed that PIN1 S4 as well as PIN3 S4 and S5 are major target sites for D6PK. This conclusion found support also in the analysis of the in vivo phosphorylation levels at these sites since phosphorylation at S4 and S5 was strongly reduced in the *d6pk012* mutant. Interestingly, the preferential D6PK phosphosites S4 and S5 are not conserved in PIN2 and it is striking that the respective domains in PIN2 carry small insertions when compared to the other PIN proteins. Thus, the activation of PINs by phosphorylation may be regulated by the presence and abundance of activating kinases such as D6PK but also by the availability and conservation of phosphosites in their PIN targets.

Besides S4 and S5, PINs must have other phosphosites that are targeted by D6PK since auxin transport defects in *pin* mutants expressing the respective PIN S4A and S5A mutant variants are partially complemented and not as severe as those observed in the *d6pk012* loss-of-function mutants. Besides phosphorylations at S1, which are not affected in *d6pk012* mutants, S2 and S3 would be other possible target sites since their mutation further impairs D6PK-dependent PIN phosphorylation in vitro and auxin transport in the oocyte system. In this respect, it is unfortunate that we were unable, for technical reasons, to measure phosphorylation at S2 and S3 in the *d6pk* and *pid* mutants.

Our study also addressed the functional role of PID and the PID-related WAG1/WAG2 kinases in the control of auxin transport. While we found that PID and WAG2 activate PIN-mediated auxin efflux in *Xenopus* oocytes, we showed at the same time that PID has different phosphosite preferences when compared to D6PK. We observed these phosphosite preferences when analyzing PIN phosphorylation at S1–S5 in in vitro phosphorylation experiments, auxin transport in oocytes, and PIN phosphorylation by quantitative mass spectrometry in the *d6pk012*, *pid* and *wag1 wag2* mutants. Whereas D6PK appeared to have a preference for the S4 and S5 sites in PIN1 and PIN3, PID

preferentially phosphorylated the previously identified S1–S3 phosphosites. S1–S3 are highly related to each other and also highly conserved among all five plasma membrane-resident PIN auxin efflux carriers including PIN2.

Although D6PK had a preference for S4 and S5 phosphorylation, our in vitro phosphorylation experiments as well as the auxin transport experiments in oocytes further suggested that the phosphorylation of S1–S3 also contributes to full PIN phosphorylation and activation by D6PK. The respective inverse observations were made with PID. Whereas PID phosphorylation and activation of PIN1 was strongly impaired when S1–S3 were mutated, full impairment of phosphorylation and activation could only be achieved after S4 mutation. In this respect, we consider the complementation experiments of the *pin1* mutant with the wild type PIN1 and the mutant PIN1 S4A transgenes particularly insightful. Here, we found that basipetal auxin transport in the inflorescence stem was partially impaired when PIN1 S1 was mutated whereas the strong inflorescence differentiation phenotype of the *pin1* mutant was rescued. The partial complementation of the *pin1* auxin transport defect indicates that PIN1 S4 is not the only phosphosite required for D6PK-dependent PIN1 activation and basipetal auxin transport. As such, this result is in agreement with the results of our in vitro phosphorylation and oocyte auxin transport experiments, which showed that D6PK can also activate PIN1 through S1–S3 phosphorylation. On the other side, the rescue of the differentiation defect can be explained because the PIN1 S4A protein still contained the preferential phosphorylation sites for PID. As shown in the in vitro phosphorylation experiment as well as in the oocyte auxin transport experiment, the PIN1 S4A mutant variant is neither strongly impaired in its phosphorylation by PID nor in its activation by PID. Thus, the essential phosphosites required for PID-dependent PIN activation and PIN polarity changes are retained in PIN1 S4A. Therefore, the selective functionality of this mutant PIN1 in the context of inflorescence development indirectly supports the findings of our other analyses.

Along the same lines, we also studied the ability of a PIN3 S4A S5A transgene to rescue the strong phototropism defect of the *pin347* triple mutant. We had previously shown that *d6pk d6pkl1* double mutants as well as *pin347* triple mutants are severely compromised in phototropic hypocotyl bending (*Willige et al., 2013*). We had shown that this phenotype could be explained by a strong defect in basipetal auxin transport and the apparent accumulation of auxin in the cotyledons of dark-grown seedlings, which, in turn, correlated with the absence of an auxin maximum in the bending zone of the hypocotyl (*Willige et al., 2013*). At the same time, PID-dependent PIN3 polarity changes could still take place in the *d6pk012* mutant indicating that PID can function independently from D6PK on PIN3. Our observation that the PIN3 S4A S5A transgene could only partially rescue the *pin347* triple mutant phenotype supports the notion that phosphorylation at these sites is important for PIN3 activation but suggests further that other phosphosites, such as S1–S3, may also be targeted by D6PK. This partial inactivation of an S4A S5A mutated PIN3 as observed *in planta* is in agreement with the partial impairment of PIN3 S4A S5A phosphorylation in in vitro phosphorylation experiments with D6PK as well as the fact that there is still a residual activation when PIN3 S4A S5A is activated with D6PK in the oocyte system. With regard to the relevance of S1–S3 for PIN3 function, we found that mutation of PIN3 S1–S3 or S1–S5 rendered this PIN3 non-functional when introduced as a transgene into *pin347*. Since these mutant PIN3 variants would be expected to be impaired in D6PK-dependent basipetal auxin transport as well as in PID-dependent PIN3 polarity changes, it is difficult based on the present depth of analysis to judge whether the non-functionality of PIN3 S1A–S3A or PIN3 S1A–S5A is primarily caused by a defect in basipetal auxin transport, a defect in changing PIN3 polarity or a combination of both.

In our experiments, PIN phosphorylation led to a direct activation of auxin efflux in oocytes. The analysis of auxin transport in Arabidopsis inflorescence stems suggested that this might indeed be the primary function of this modification since auxin transport was strongly impaired in *d6pk* loss-of-function mutants while PIN abundance at the plasma membrane was not altered. This observation does, however, not rule out that PIN phosphorylation has other regulatory roles such control of PIN polarity by PID or WAG-dependent phosphorylation (*Friml et al., 2004*; *Dhonukshe et al., 2010*). Changes in PIN polarity as they are observed after *PID* or *WAG2* overexpression are not observed after *D6PK* overexpression (*Zourelidou et al., 2009*; *Dhonukshe et al., 2010*; *Barbosa et al., 2014*). The differential effect of D6PK and PID/WAGs on PIN may have its molecular basis in the differential phosphosite preferences of the two kinases. Common to both kinases seems, however, the fact that their phosphorylation activity is antagonistically regulated by phosphatases. While the phenotypic effects of PID can be antagonized by PP2A phosphatases (*Michniewicz et al., 2007*), removal of the

D6PK from the plasma membrane through BFA treatment resulted in an almost immediate decrease in PIN1 phosphorylation. Thus, it can be speculated that also D6PK-dependent PIN phosphorylation is antagonized by phosphatases, the identities of which remain to be determined.

Our data also suggest that PIN1 and PIN3 phosphorylation is not only controlled by the presence of D6PK at the plasma membrane but also by auxin itself. Using quantitative SRM analyses, we could show that PIN1 S4 as well as PIN3 S4 and S5 phosphorylation increases in response to auxin treatment in the wild type. In the *d6pk012* mutant, the loss of phosphorylation at these preferential D6PK phosphosites could not be compensated by auxin application suggesting that these auxin-dependent phosphorylations are D6PK-dependent and may be mediated either directly by D6PK or by D6PK acting as an indirect auxiliary factor. Although we observed a minor increase in PIN phosphorylation at the D6PK phosphosites in the *d6pk012* triple mutant, these increases were comparatively minor and may be attributed to phosphorylation through D6PKL3, which is still expressed in the *d6pk012* mutant. Alternatively, they may be attributed to the activity of other PIN-regulatory kinases such as the PID/WAGs or other as yet uncharacterized protein kinases. Theoretically, it could be envisioned that the auxin-dependent increases in PIN phosphorylation are the consequence of the previously reported inhibitory effects of auxin on PIN endocytosis (*Paciorek et al., 2005*). In this case, PIN phosphorylating kinases would encounter their PIN targets simply for a longer period of time thereby increasing the chances for phosphorylation. Unraveling the identity of the underlying auxin-sensory mechanism and its molecular details will be an interesting avenue for future investigations (*Dharmasiri et al., 2005*; *Parry et al., 2009*; *Robert et al., 2010*).

We recently reported that auxin treatment led to a transient dissociation of D6PK from the plasma membrane in root cells There, this auxin response correlates with a slight decrease in PIN1 phosphorylation as judged by immunoblots (*Barbosa et al., 2014*). In contrast, we report here that auxin promotes PIN phosphorylation in inflorescence stems as determined by quantitative mass spectrometry of PIN1 S4, PIN3 S4 and PIN3 S5. It is at present difficult for us to reconcile these two apparently contrasting observations. We can therefore only argue that PIN phosphorylation is controlled by different auxin-dependent regulatory mechanisms in different tissues.

In summary, our study provides evidence that PIN-mediated auxin efflux requires activation by PIN phosphorylating kinases such as D6PK and PID/WAGs. Several of our findings point at a differential biochemical activity of these two AGCVIII kinase representatives on PINs that may explain their differential effects in controlling PIN polarity, auxin transport, and plant growth. The differential PIN-dependent distribution of auxin within the plant is of pivotal importance for the regulation of a multitude of processes in plant growth and development. It is our view that the activation of PINs by D6PKs and PID/WAGs is a crucial component of the control of auxin transport that must be taken into account to understand auxin transport within the plant and to ultimately understand plant growth.

## Materials and methods

### Biological material

The following mutant alleles were used for this study: Single, double and triple mutants of *d6pk-1* (*d6pk0*; SALK_061847), *d6pkl1-1* (*d6pk1*; SALK_056618), *d6pkl2-2* (*d6pk2*; SALK_086127) (*Zourelidou et al., 2009*); *d6pk012* triple mutants with a complementing *D6PKp:YFP:D6PK* transgene (*Willige et al., 2013*); DR5:GFP (*Jonsson et al., 2006*); *pid-14* (SALK_049736); *pin1* (SALK_047613); *pin3-3 pin4-101 pin7-102* (*Willige et al., 2013*); PIN1:GFP (*Grieneisen et al., 2007*); *wag1 wag2* (*Santner and Watson, 2006*). *D6PKp:GUS* transgenic lines expressing the ß-glucuronidase (GUS) reporter under control of *D6PK* promoter (*D6PKp*) fragments as previously described (*Zourelidou et al., 2009*). *PIN1p:GUS*, *PIN3p:GUS*, *PIN4p:GUS* and *PIN7p:GUS* (*Vieten et al., 2005*) were obtained from the Nottingham Arabidopsis Stock Center (NASC) or are a gift from Christian Luschnig (Vienna, Austria).

### Cloning procedures

Primer sequences for cloning, insert amplification and site-directed mutagenesis are listed in *Supplementary file 1A*.

For in vitro transcription prior to protein translation in oocytes, *PIN1* and *PIN3* were inserted into the expression vector pOO2 (*Broer, 2010*). To this end, the genes were amplified from cDNA templates and cloned as blunt-ended Phusion polymerase-amplified (Biozym, Hessisch Oldendorf, Germany) PCR fragments into the *Sma*I or *Eco*RV site of pOO2. The S to A mutations were introduced

by PCR-based site-directed mutagenesis using primers carrying mutations for the respective S to A replacements. *YFP:D6PK* and kinase-dead *YFP:D6PKin* were amplified in a similar manner as described for *PIN1* and *PIN3* from previously described vector templates (*Zourelidou et al., 2009*) and inserted into the *Eco*RV site of pOO2. A PCR-fragment of the *PID* coding sequence was first inserted into pJET1.2 (Fisher Scientific, Schwerte, Germany) and from there transferred as an *Xho*1/*Xba*1 fragment into pOO2. A PCR-fragment of the *WAG2* CDS was cloned directly into pOO2 after *Xba*1/*Nco*1 digestion. The *phot1* CDS was cloned as a *Bam*H1/*Pst*1-digested PCR fragment into the corresponding sites of pOO2.

Constructs for the expression and purification of glutathione-S-transferase (GST)-tagged PIN and D6PK were previously described (*Zourelidou et al., 2009*; *Huang et al., 2010*; *Willige et al., 2013*). GST:PID was obtained by inserting a Gateway-compatible PCR-fragment obtained from a *PID* cDNA with the primers PID-GW-FW and PID-GW-RV into pDONR201 before transferring the *PID* insert to pDEST15 (Life Technologies, Carlsbad, CA).

*D6PKp:YFP:D6PK* was generated by inserting YFP:D6PK including a terminator sequence as an *Xho*I/*Not*I fragment excised from pEXTAG-YFP-GW[1] into pGREEN0229[12] to generate pGREEN-YFP:D6PK. A 1977 bp *D6PK* promoter-PCR fragment (*D6PKp*) was cloned into pCR2.1 (LifeTechnologies, Carlsbad, CA) and subsequently inserted as *Kpn*I/*Xho*I fragment into pGREEN-YFP:D6PK to generate *D6PKp:YFP:D6PK*. To obtain *D6PKp:D6PK*, the *D6PK* CDS plus terminator sequence was excised from p35SGW-MYC as an *Xho*1/*Spe*1 fragment (*Zourelidou et al., 2009*) and inserted into the respective sites of pGREEN0229 containing *D6PKp*. In a similar manner, the *PID* promoter (2344 bp) was amplified by PCR from genomic DNA, inserted into pCR2.1 (Life Technologies, Carlsbad, CA) and from there transferred as a *Kpn*1/*Xho*1 fragment into pGREEN0229. To obtain D6PKp:PID and PIDp:PID, the *PID* CDS was inserted as a Gateway-technology compatible insert into p35SGW-MYC and from there cloned as an *Xho*1/*Spe*1 insert into pGREEN0229 containing the *D6PKp* or *PIDp*.

Genomic *PIN1* constructs were prepared by insertion of a 3558 bp *Sal*1/*Not*1-digested PCR fragment including the *PIN1* open reading frame and terminator into pGREEN0229. Subsequently, a 2081 bp *PIN1* promoter fragment was inserted upstream from *PIN1* as a *Kpn*1/*Sal*1-digested PCR fragment. Mutations for the S4A replacement were introduced by site-directed mutagenesis (*Sawano and Miyawaki, 2000*). The constructs were transformed into heterozygous *PIN1/pin1* (SALK_047613) plants by Agrobacterium-mediated transformation[13] and *pin1* homozygous lines carrying the *PIN1* transgenes were isolated from the progeny. Plants expressing comparable levels of the PIN1 protein were identified by immunoblotting.

Constructs for the expression of *PIN3* under the control of its own promoter were obtained by amplifying a fragment spanning the region from 1776 bp upstream of the *PIN3* translation start site to 621 bp downstream of the *PIN3* stop codon with the primers PIN3g-ApaI-FW and PIN3g-NotI-RV and inserted into the *Kpn*I and *Not*I sites of pGREEN0229. The S1A through S5A mutations were introduced into the wild type construct by PCR-based site-directed mutagenesis using the primers listed in *Supplementary file 1A*. The constructs were transformed by Agrobacterium-mediated transformation into *pin3 pin4 pin7* mutants (*Willige et al., 2013*) and phototropism experiments were performed on T2 progeny seedlings segregating for the *PIN3* transgenes in the *pin3 pin4 pin7* mutant background. Assuming that 25% of the segregating population represent non-transgenic *pin3 pin4 pin7* segregants, the 25% of the seedlings of the analysed population (n >50 for *pin3 pin4 pin7* PIN3 S4A S5A; n >25 for *pin3 pin4 pin7* PIN3 S4A S5A and *pin3 pin4 pin7* PIN3 S1A S2A S3A S4A S5A) with the lowest hypocotyl angle were excluded from the analysis. The T2 progeny of at least three independent transgenic lines was analysed for each transgene, and in each case the three lines with the strongest phenotypic suppression were chosen for the graphic representations and statistical analyses. The variance between the individual transgenic populations was analysed with a Kruskal–Wallis ANOVA on ranks (*Kruskal and Wallis, 1952*).

## Arabidopsis auxin transport assays

To measure auxin transport in Arabidopsis inflorescence stems, 25-mm stem sections were cut above the rosette of 5-week-old plants and placed, in inverted orientation, into 30 µl auxin transport buffer containing 500 pM IAA, 1% (wt/vol) sucrose, 5 mM 2-(N-morpholino)ethanesulfonic acid (MES), [pH 5.5] with or without 100 µM 1-N-naphthylphthalamic acid (NPA). At the beginning of the transport experiment, the stem segments were transferred to 30 µl auxin transport buffer containing 417 nM (11 kBq) [3H]-IAA (American Radiolabeled Chemicals, St. Louis, MO). After 2 hr, 5-mm segments

were dissected from the inflorescence stem, the lowermost 5-mm segment was discarded, and the remaining segments were macerated overnight in 3 ml QuickSafe A (Zinsser Analytic, Frankfurt, Germany). [$^3$H]-IAA was quantified using a liquid scintillation analyzer (Tri Carb 2100TR; Perkin–Elmer). The results presented are the average and standard deviation of at least four biological replicate measurements in the case of wild type, *d6pk* mutants, *pin1 PIN1* and *pin1 PIN1S4A*, and at least two biological replicates in the case of *pin1* or the NPA-treated wild type. The experiments were repeated with comparable results and the result of a comparable experiment is shown. Where relevant, auxin transport measurements were compared using the linear mixed-effects model analysis (fixed factors) using the R software package.

## Oocyte auxin efflux assays

*Xenopus laevis* oocyte collection was performed as previously described and cRNA injection was carried out the day after surgery (**Kottra et al., 2009**). cRNA was synthesized using the mMessage Machine SP6 Kit (Life Technologies, Carlsbad, CA) and cRNA concentration was adjusted to 300 ng/µl PIN and 150 ng/µl protein kinase, respectively. Oocytes were injected with ~50 nl of a 1:1 mixture of cRNAs for PIN and protein kinase. If only PIN or protein kinase cRNA was injected, the cRNA was mixed 1:1 with water (mock control). Following injection, oocytes were incubated in Barth's solution containing 88 mM NaCl, 1 mM KCl, 0.8 mM MgSO$_4$, 0.4 mM CaCl$_2$, 0.3 mM Ca(NO$_3$)$_2$, 2.4 mM NaHCO$_3$, 10 mM HEPES (pH 7.4) supplemented with 50 µl gentamycine at 16°C for 4 days to allow for protein synthesis. An outside medium buffer at pH 7.4 was chosen to prevent passive rediffusion of IAA into the oocytes, which would take place at acidic pH. At the beginning of the oocyte experiment, 10 oocytes were injected for each time point with 50 nl of a 1:5 dilution (in Barth's solution) of [$^3$H]-IAA, 25 Ci/mmol; 1 mCi/ml (ARC, St. Louis, MO) to reach an intracellular oocyte concentration of ~1 µM [$^3$H]-IAA based on an estimated oocyte volume of 400 nl (**Broer, 2010**). After [$^3$H]-IAA injection, oocytes were placed in ice-cold Barth's solution for 10 min to allow substrate diffusion and closure of the injection spot. Subsequently, oocytes were washed and transferred to Barth's solution at 21°C to allow for auxin efflux. To stop auxin efflux, oocytes were washed twice and lysed individually in 100 µl 10% SDS (wt/vol) at selected time points and the residual amount of [$^3$H]-IAA in each oocyte was determined by liquid scintillation counting. At least 10 oocytes were measured per time point and mock as well as other negative controls were performed with the same oocyte batch to account for differences between batches. The relative transport rates of an experiment were determined by linear regression as shown in **Figure 5—figure supplement 1**. Transport rates of different biological replicates (i.e. oocytes collected from different female donors) were averaged and are presented as mean and standard error of at least three biological replicates. Comparability in protein expression between the respective wild type and mutant protein variants and between the experiments was confirmed using immunoblots or confocal laser scanning microscopy with a Axiovert 200 M microscope equipped with a LSM 510 META laser scanning unit (Zeiss, Jena, Germany).

## Immunoblots

For protein extraction from *Xenopus laevis* oocytes, up to 25 oocytes were homogenized by trituration on ice in a homogenization buffer containing 50 mM Tris–HCl, 100 mM NaCl, 1 mM EDTA, 1 mM Pefabloc (400 µl/oocyte). The homogenate was centrifuged at 2000×*g* for 10 min at 4°C and the supernatant was transferred to a polyallomer microfuge tube (Beckman Instruments, Fullerton, CA). Membrane proteins were pelleted from this supernatant at 150,000 *g* for 30 min at 4°C. The supernatant (CF, cytoplasmic fraction) was recovered and the pellet (MF, microsomal membrane fraction) was resuspended in homogenization buffer supplemented with 4% (wt/vol) SDS (8 µl per oocyte). The equivalent of 1/16th oocyte was loaded for immunoblots. To detect PIN phosphorylation, the homogenization buffer was supplemented with PhosSTOP phosphatase inhibitor cocktail (Roche, Penzberg, Germany) and the samples were immediately subjected to immunoblot analysis with the following antisera: anti-PIN1 (1:5000; NASC), anti-PIN3 (1:3000; NASC), anti-GFP (1:2000; Life Technologies, Carlsbad, CA), anti-UCN (1:2000; a gift from Kay Schneitz, Technische Universität München, Germany [**Enugutti et al., 2012**]), anti-phot1 and anti-S851-phot1 antisera (1:1000; a gift from Shin-ichiro Inoue, Nagoya University and Ken-ichiro Shimazaki, Kyushu University, Japan [**Inoue et al., 2008**]). Secondary detection was performed with donkey anti-sheep IgG-HRP (1:5000; Dianova, Hamburg, Germany) and goat anti-rabbit IgG-HRP (1:5000; Santa Cruz Biotechnology, Santa Cruz, CA). PIN1 western blots from PIN1 transgenic plants were performed as previously described (**Willige et al., 2011**).

## GUS staining

For GUS staining, inflorescence stem segments were sectioned with a razor blade, fixed for 15 min in heptane and stained for 4 hr or overnight (*PIN7p:GUS* only) with GUS-staining solution (100 mM Na-phosphate buffer pH 7.0, 0.1% (vol/vol) Triton X-100, 0.2 mg/ml 5-bromo-4-chloro-3-indolyl β-D-glucuronic acid) and subsequently destained in 70% (vol/vol) ethanol. Images were taken with a Leica MZ16 microscope (Leica Microsystems, Heerbrugg, Switzerland).

## In vitro phosphorylation experiments

Peptides for phosphorylation experiments as listed in *Supplementary file 1B* were synthesized by standard automated solid phase chemistry following the Fmoc (Fluorenylmethyloxycarbonyl) strategy (Multipep, Intavis, Cologne). Phosphorylation experiments were performed using 0.5 μg purified GST:D6PK and 50 μM synthetic peptide in a reaction buffer containing 125 mM Tris pH 7.5, 25 mM $MgCl_2$, 1 mM EDTA, 1 × Complete protease inhibitor cocktail (Roche, Penzberg, Germany), 0.09 mM ATP, 0.0125% xylene cyanol and 0.1 μl [©−$^{32}$P]ATP (370 MBq, specific activity 185 TBq; Hartmann Analytic, Braunschweig, Germany). The reactions were incubated for 1 hr at 30°C and 2 μl of the 20 μl reaction were spotted in duplicates on P81 ion exchange chromatography paper (GE Healthcare, Freiburg, Germany). Air-dried chromatography papers were washed with 0.85% phosphoric acid, dried and exposed to X-ray film.

Phosphorylation experiments with recombinant PIN cytoplasmic loop substrates were performed using 0.2 μg GST:D6PK or GST:PID and 0.5 μg GST:PIN substrate in a reaction buffer containing 25 mM Tris pH 7.5, 5 mM $MgCl_2$, 0.2 mM EDTA, 1 × cOmplete protease inhibitor cocktail (Roche, Penzberg, Germany), and 0.5 μCi [©−$^{32}$P]ATP (370 MBq, specific activity 185 TBq; Hartmann Analytic, Braunschweig, Germany). Reactions were incubated for 1 hr at 30°C and separated on 10% SDS-PAGE. Gels were dried using a vacuum drier and exposed to X-ray film. Band intensities were quantified using MultiGauge v.3.0 and normalized to the band intensities of the wild type.

Phosphorylation experiments with recombinant PIN cytoplasmic loop substrates for mass spectrometric analysis were performed at 30°C for 1 hr in a non-radioactive reaction buffer containing 25 mM Tris pH 7.5, 5 mM $MgCl_2$, 0.2 mM EDTA, 1 × cOmplete protease inhibitor cocktail (Roche, Penzberg, Germany), 0.15 mM ATP, 1 × PhosSTOP phosphatase inhibitor cocktail (Roche, Penzberg, Germany) with 5 μg purified recombinant D6PK and 5 μg purified recombinant PIN cytosolic loop. For subsequent mass spectrometric analyses, the reactions were separated on a 10% SDS-PAGE and stained with Coomassie Brilliant Blue. PIN bands were cut from the gel, destained with two washes of $H_2O$ and two washes of 50% acetonitrile/50 mM $NH_4HCO_3$ pH 8 at 37°C. The bands were then sliced into small pieces (1 mm$^2$) and transferred to a low binding microcentrifuge tube. The gel pieces were then covered in a solution with 50 mM dithiothreitol (DTT), 50 mM $NH_4HCO_3$ and incubated for 1 hr at 60°C. After cooling to room temperature, the solution was replaced by 100 mM iodoacetamide in 50 mM $NH_4HCO_3$ and incubated for at least 1 hr in the dark. Subsequently, the gel pieces were washed three times by vortexing for 10 min in 50 mM $NH_4HCO_3$, pH 8. Following removal of the wash solution, the gel pieces were dried in a SpeedVac concentrator for 30 min and then incubated overnight in 10 μl Bovine Sequencing Grade Trypsin (Roche, Penzberg, Germany) dissolved in 50 mM $NH_4HCO_3$, 1 mM $CaCl_2$. The trypsin solution was subsequently removed and transferred to a low binding tube. 10 μl of trifluoroacetic acid (TFA; 5% wt/vol $H_2O$) were then added to the gel pieces and after sonication for 1 min the supernatant was transferred to the tube containing the previous liquid. The same procedure was repeated by adding 10 μl 15% acetonitrile/1% TFA to the gel pieces and combining the liquid with the previous supernatants. Mass spectrometry was performed using an nLC-LTQ-Orbitrap tandem mass spectrometer at Biqualys (Wageningen, The Netherlands), and the data were analysed using the Bioworks software (Thermo Fisher Scientific, Ulm, Germany).

## Protein alignment

PIN protein alignments was performed using the ClustalW alignment option of the Geneious (Biomatters, Auckland, New Zealand) software package.

## Phototropism experiments

Seedlings were grown in the dark at 22°C on vertically oriented half-strength Murashige and Skoog (MS) agar (0.8%) plates for 3 to 4 days. Agravitropically growing seedlings were reoriented toward the

gravity vector in safe green light 2 to 4 hr before the experiment. The seedlings were then transferred to GroBank growth chambers (CLF Plant Climatics, Wertingen, Germany) and illuminated with unilateral white light (100 μmol m$^{-2}$ s$^{-1}$). Plates were subsequently scanned and hypocotyl bending was measured for each seedling using the NIH ImageJ software.

## Immunostaining and confocal microscopy

The rabbit anti-PIN1 pS4 antibody was generated with the phosphorylated synthetic peptide SGGGRN-S(PO$_3$H$_2$)-NFGPGE followed by affinity-purifications against the non-phosphorylated and phosphorylated peptide at Eurogentec (Liege, Belgium). Immunostaining was performed on roots of 5-day-old seedlings as previously described (*Sauer et al., 2006*) using rabbit anti-S4-P (dilution 1:300), goat anti-PIN1 (1:400; NASC Nottingham, UK), and mouse anti-GFP (1:300; Roche, Penzberg, Germany), and as secondary antibodies anti-rabbit Cy3 (1:500; Dianova, Hamburg, Germany), anti-goat FITC (1:100; Dianova, Hamburg, Germany), anti-rabbit FITC (1:300; Dianova, Hamburg, Germany) as well as anti-mouse ALEXA FLUOR 488 (1:500; LifeTechnologies, Carlsbad, CA). For BFA treatment, seedlings were immersed in BFA [50 μM]-containing liquid MS medium prior to analysis. All images were taken with an Olympus FV1000 confocal microscope (Olympus, Hamburg, Germany). The experiment was repeated several times with reproducible outcome, representative images are shown.

Live imaging of DR5:GFP or fluorescent protein-tagged proteins was performed as previously described (*Barbosa et al., 2014*).

## SRM analysis

Protein extracts for SRM analyses were prepared from 5 cm primary inflorescence stem segments, excised at the base of the infloresence stems, from 5 week-old Arabidopsis plants grown in continuous light. Total protein extracts were prepared in an extraction buffer containing 50 mM Tris–HCl pH 7.5, 150 mM NaCl, 0.5% Triton X-100, 0.1 mM MG132 (Z-Leu-Leu-Leu-al), 1 mM PMSF (phenylmethylsulfonyl fluoride), protease inhibitor cocktail (Sigma-Aldrich, St. Louis, MO) and PhosSTOP (Roche, Mannheim, Germany). From each sample, 150 μg total protein were prepared in 100 μl extraction buffer and precipitated with 10 ng/μl glycogen, 400 μl ethanol (HPLC Gradient Grade, Roth, Karlsruhe, Germany), 25 mM NaOAc 2.5 M pH5.2 for 4 hr at room temperature. Subsequently, the samples were centrifuged at 10,000×*g* and air-dried for subsequent SRM analysis.

Protein pellets were resuspended in 6 M urea, 2 M thiourea, pH 8. Protein disulfide bridges were reduced by adding DTT and free cysteine residues were subsequently alkylated using iodacetamide. 150 μg protein was then digested using sequencing grade trypsin (Promega) and desalted over C18 tips[22]. Phosphopeptides were enriched over titanium dioxide[23] and eluted phosphopeptides as well as flow-through after peptide binding to titanium dioxide were kept for analysis. Synthetic peptides with fully $^{13}$C and $^{15}$N-labeled C-terminal K or R were synthesized (Thermo Fisher Scientific, Ulm, Germany; *Supplementary file 1C*) and spiked into the tryptic peptide mixture at concentrations ranging from 40 to 250 fmol depending on peptide ionization properties.

Tryptic peptide mixtures including heavy standard peptides were then analysed by SRM using nanoflow HPLC (Easy nLC, Thermo Scientific, Ulm, Germany) coupled to a triple quadrupole mass spectrometer as mass analyser (TSQ Quantum Discovery Max, Thermo Scientific, Ulm, Germany). Peptides were eluted from a 75 μm analytical column (Easy Columns, Thermo Scientific, Ulm, Germany) on a linear gradient running from 10% to 30% acetonitrile in 60 min and were ionized by electrospray directly into the mass spectrometer. Specifically, phosphorylated and non-phosphorylated peptides were selected as targets of analysis after optimization of ionization conditions using the standard peptides. Visible transitions were selected from acquired mass spectra of the synthetic standard peptides. A list of transitions used for each (phospho)peptide sequence is available as *Supplementary file 1C*. The quadrupole Q1 was set as a mass filter for the respective parent ion, while Q3 was set to monitor specific fragment ions. For each peptide, at least three fragment ions were used. Mass width for Q1 and Q3 was 0.7 Da, scan time 5 ms.

Data analysis involving merging of fragment ion information to a parent ion sum of intensities and calculation of peak areas was done using the Software Pinpoint v.1.0 (Thermo Scientific, Ulm, Germany). For quantitative analysis of peptide abundance, ion intensity sums of the measured transitions were used and averaged between up to three biological replicates. Ion intensity sums of spiked-in heavy peptide were used to normalize for sample-to-sample variation.

## Acknowledgements

The authors are grateful to Helene Prunkl and Hannelore Daniel (Technische Universität München, Germany) for supplying *Xenopus laevis* oocytes. The authors wish to thank various colleagues for sharing antibodies, Siv Ahlers, Manuel Jeller, Susanne Weber, and Mon Yinnavong (Technische Universität München, Germany) for providing materials, and Manuela Bog (Universität Regensburg, Germany) for help with statistical analyses. This work was supported by grants from the Deutsche Forschungsgemeinschaft (grant numbers SCHW751/8-1 and SFB924) to CS and UZH (SFB924) and by a fellowship from the Fundação para a Ciência e a Tecnologia to ICRB.

## Additional information

### Funding

| Funder | Grant reference number | Author |
|---|---|---|
| Deutsche Forschungsgemeinschaft (DFG) | SCHW751/8-1 | Claus Schwechheimer |
| Deutsche Forschungsgemeinschaft (DFG) | SFB924 | Ulrich Z Hammes |

The funders had no role in study design, data collection and interpretation, or the decision to submit the work for publication.

### Author contributions

MZ, BA, BW, BCW, AF, VS, SAP, JC, SFB, WXS, Conception and design, Acquisition of data, Analysis and interpretation of data; ICRB, Conception and design, Analysis and interpretation of data, Drafting or revising the article, Contributed unpublished essential data or reagents; HH, Conception and design, Acquisition of data; BK, Conception and design, Acquisition of data, Contributed unpublished essential data or reagents; UZH, Conception and design, Acquisition of data, Analysis and interpretation of data, Drafting or revising the article; CS, Conception and design, Analysis and interpretation of data, Drafting or revising the article

### Ethics

Animal experimentation: This study was performed in strict accordance with the recommendations and guidelines based on the Tierschutzgesetz (TierSchG) of the Federal Republic of Germany.

## Additional files

### Supplementary file

• Supplementary file 1. (**A**) Primer sequences used in this study. (**B**) Sequences of peptides used for phosphorylation experiments. (**C**) Peptides used for SRM analyses.

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
