## [Decision Letter]

Thank you for sending your work entitled “Auxin efflux by PIN-FORMED proteins is activated by D6 PROTEIN KINASE and PINOID” for consideration at *eLife.* Your article has been favorably evaluated by Detlef Weigel (Senior editor) and 2 reviewers.

The Reviewing editor and the other reviewers discussed their comments before we reached this decision, and the Reviewing editor has assembled the following comments to help you prepare a revised submission. We would like to ask you to pay particular attention to the comments regarding the apparent disconnect between the macroscopic phenotype of pin mutants complemented by phosphosite variants and the importance of those sites in the transport assays.

Auxin efflux carriers of the PIN-FORMED (PIN) family are the principal determinants of polar auxin transport in plant development. In this paper, you present a detailed account on the regulation of their activity through differential phosphorylation. There is conclusive evidence that PIN proteins are phosphorylated by PINOID (PID) and related (WAG) kinases as well as D6 protein kinases (D6PKs) at specific sites. You show that these phosphorylations not only affect, as previously shown, the polarity of PINs (i.e. through PID/WAG), but also plainly the capacity of PINs to transport auxin across the plasma membrane. These findings fit with your previous molecular genetic analyses of D6PKs and force us to rethink the notion that auxin flux can be solely predicted from PIN localization, an assumption that underlies various models in this area of investigation.

There are a few suggestions that could improve this otherwise high quality manuscript:

1) Given that you provide evidence for a role of D6PKs in hypocotyl phototropism, could you please also provide images of D6PK reporter construct expression in hypocotyl(s) (and) cells?

2) The biochemical part and auxin efflux assay in oocytes seems very well done to me. Still, one is wondering whether you could provide raw data of the phosphoproteomic analysis (i.e. what peptides where found, how many, etc.)?

3) There is one genetic experiment that the reviewers had a hard time interpreting: the PIN1S4A variant can rescue a *pin1* null mutant, yet basipetal transport is compromised. However, from a phenotypic perspective, one could argue then that S4 does not matter, the plants look wild type. This disconnect between biochemical/physiological results and genetic results merits some more attention. Given the strong *pin1* phenotype, one would expect at least some type of intermediate phenotype in the lines complemented by the S4A variant if this site matters. Did you try to complement the mutant with a PIN1 S4A D215A double point mutant and if so, did this rescue only partially?

4) Also with respect to the above, can PIN3 expressed under PIN1 promoter rescue the *pin1* mutant, and if so, what about the PIN3 S4A S5A variant?

5) Finally, was the *pin1* mutant used a true null allele, and could you quantify the rescue in terms of, e.g., perfect floral meristems or overall branch number to eventually detect intermediate phenotypes?

6) With regard to complementation again, the reviewers had a hard time finding a clear description of *pin347* triple mutants in the literature. You focus on this triple mutant's phototropism defect in your assays, but are there other phenotypes that could be monitored?

7) Given the speed of the phosphorylation induced by auxin, it seems surprising that it is supposedly under control of the TIR1/AFB auxin receptors. Does this suggest that these receptors have a non-transcriptional activity in this? Then, does auxin-induced phosphorylation still occur when transcription is blocked?

8) Has it been addressed whether S4 and S5 phosphorylation are specific to activity changes of PINs or whether they also play a role in localization? Finally, do they affect the kinetics of endocytic PIN recycling?

9) Figure 7 and accompanying text: if there were no difference in total abundance of PIN proteins in the *dp6k* mutants, shouldn't the sum of PIN_pS4 and PIN_S4 be the same in WT and mutants?

10) Figure 8 and accompanying text: if the S4 of PIN1 is not a target of PID/WAG kinase function, why does the S4A mutation further impair activation by PID? (compare PIN1_S1A-S3A ^+/-^ PID with PIN1_S1A-S4A ^+/-^ PID)

11)The reviewers are wondering whether it is true that Ler and Col behave differently from each other and that *abp1-5* behaves differently from Ler: it seems that in all genotypes, the amount of PIN1_pS4 doubles in the presence of auxin.

[Editors' note: further clarifications were requested prior to acceptance, as described below.]

Thank you for resubmitting your work entitled “Auxin efflux by PIN-FORMED proteins is activated by D6 PROTEIN KINASE and PINOID” for further consideration at *eLife*. Your revised article has been favorably evaluated by Detlef Weigel (Senior editor) and two reviewers. The manuscript has been improved but there are some remaining issues that need to be addressed before acceptance, as outlined below:

On a side note, you might consider to remove the part describing the TIR1 pathway-dependence of PIN phosphorylation. Although, as you indicate, the data are solid, they remain unexplored with respect to the mechanism and they might distract from the main message of the paper. However, let us emphasize that it is your decision whether to keep those data in or not.

More importantly, an issue was arising regarding genetic background. In your assays with the *abp1* mutant, you were using the weak *abp1-5* allele, and it appears that you match it to a Ler wild type control. However, the originally described *abp1-5* (Xu et al., Cell, 2010; Robert et al., Cell, 2010) was a TILLING allele obtained in the Col-0 background and cross backed to Col-0 four times. Could you please clarify the origin and nature of your allele? Finally, if the allele you used should be the described one in Col-0 background, does this change the interpretation of the results?

---

## [Author Response]

*Auxin efflux carriers of the PIN-FORMED (PIN) family are the principal determinants of polar auxin transport in plant development. In this paper, you present a detailed account on the regulation of their activity through differential phosphorylation. There is conclusive evidence that PIN proteins are phosphorylated by PINOID (PID) and related (WAG) kinases as well as D6 protein kinases (D6PKs) at specific sites. You show that these phosphorylations not only affect, as previously shown, the polarity of PINs (i.e. through PID/WAG), but also plainly the capacity of PINs to transport auxin across the plasma membrane. These findings fit with your previous molecular genetic analyses of D6PKs and force us to rethink the notion that auxin flux can be solely predicted from PIN localization, an assumption that underlies various models in this area of investigation*.

There are a few suggestions that could improve this otherwise high quality manuscript:

1) Given that you provide evidence for a role of D6PKs in hypocotyl phototropism, could you please also provide images of D6PK reporter construct expression in hypocotyl(s) (and) cells?

The role of D6PKs in phototropic hypocotyl bending has been the topic of a previous publication (Willige et al., Plant Cell, 2013). In this publication, we describe and analyze at the molecular level the non-phototropic phenotype of *d6pk* mutants as well as the non-phototropic phenotype of *pin3 pin4 pin7* mutants. In Supplemental Figure 2 of this paper, we show using promoter:GUS lines that all four *D6PK* genes are expressed in the hypocotyl.

We noted, however, based on this and other reviewer comments that we may not have sufficiently introduced and discussed the aspects of this paper related to phototropism. We have therefore added a paragraph in the Discussion to elaborate on the role of D6PKs and PID/WAGs in the control of auxin transport during phototropic hypocotyl bending. We have also made additional changes to the text to make this part of our manuscript easier to understand in the absence of detailed knowledge of the [37] paper.

*2) The biochemical part and auxin efflux assay in oocytes seems very well done to me. Still, one is wondering whether you could provide raw data of the phosphoproteomic analysis (i.e. what peptides where found, how many, etc*.*)?*

We have now added two additional data files with the requested information (Figure 3–figure supplements 1 and 2). As can be seen from these two supplementary files, almost full coverage was achieved in these mass spectrometric analyses and phosphorylations at the sites S1–S5 were observed in almost all cases for all PINs and later confirmed for all PINs, PIN1–PIN4 and PIN7 in the peptide phosphorylation experiments presented in Figure 3.

*3) There is one genetic experiment that the reviewers had a hard time interpreting: the PIN1S4A variant can rescue a* pin1 *null mutant, yet basipetal transport is compromised. However, from a phenotypic perspective, one could argue then that S4 does not matter, the plants look wild type. This disconnect between biochemical/physiological results and genetic results merits some more attention. Given the strong* pin1 *phenotype, one would expect at least some type of intermediate phenotype in the lines complemented by the S4A variant if this site matters. Did you try to complement the mutant with a PIN1 S4A D215A double point mutant and if so, did this rescue only partially?*

As stated above, we noted based on this comment that we have not paid sufficient attention to discuss in detail the results from the complementation experiments (*pin1* mutant rescue experiment; phototropism experiment with PIN3 in *pin3 pin4 pin7* mutants). For the revised manuscript, we have paid particular attention to providing a better introduction into the roles of D6PK and PID in controlling PINs in the context of shoot differentiation and auxin transport in the shoot as well as their roles in the control of phototropism.

The key to understanding these data is to realize that D6PK and PID do not regulate auxin transport in an additive manner but control distinct processes in the inflorescence and during phototropism. In brief, *pin1* mutants are defective in basipetal auxin transport in the stem and they are unable to form local auxin maxima required for lateral organ differentiation. The latter process requires a PIN1 polarity change that is mediated by PIN1 phosphorylation at the preferential PID phosphosites S1–S3. In the PIN1 S4A mutant, the phosphorylation at the S1–S3 phosphosites can still take place and, therefore, PIN1 polarity changes can still occur, local auxin maxima can form and the *pin1* mutant phenotype is rescued. At the same time, we find that *pin1* with the PIN1 S4A transgene is impaired in auxin transport. This finding is in line with the observations in the oocyte experiment where mutation of S4 also partially impairs auxin efflux. According to the oocyte experiments, additional mutations of S1 – S3 would be needed to fully impair basipetal auxin transport but such an experiment would not allow us judging the importance of the previously uncharacterized S4 phosphosite.

The results obtained with the PIN3 S4A S5A transgene can be interpreted in a similar manner. The oocyte experiments would predict that this mutant variant is only partially impaired in its activation by D6PK and basipetal auxin transport. The PIN3 S1A – S3A mutations affect PID-dependent PIN3 polarity changes, which are essential for lateralization of auxin and possibly also for PID-dependent activation of PIN3 auxin transport and they are therefore more severe. Goal of this particular experiment was to demonstrate the functional importance of the novel S4 and S5 phosphosites of PIN3.

*4) Also with respect to the above, can PIN3 expressed under PIN1 promoter rescue the* pin1 *mutant, and if so, what about the PIN3 S4A S5A variant?*

This is in fact an interesting experiment, which we have however not conducted yet. When designing the already very complex set of experiments on PIN phosphorylation presented in this paper, we were very much interested to be able to provide an as complete, as in complementary, set of data for selected phosphosites. Already the analysis of four or five different phosphosites of a single PIN gives rise to 24 or even 120 possible combinations of phosphosite mutations. At least for us, it is impossible to evaluate such a large number of mutants in three different assay systems (in vitro phosphorylation, oocyte experiments, and mutant complementation). We have therefore decided after the initial characterization of the four and five phosphosites in PIN1 and PIN3, respectively, to concentrate on aspects of S4 and S5 in the context of the native proteins. We ask for your understanding.

*5) Finally, was the* pin1 *mutant used a true null allele, and could you quantify the rescue in terms of, e.g., perfect floral meristems or overall branch number to eventually detect intermediate phenotypes?*

Yes, the *pin1* mutant used in our studies is a true null. The mutant displays all phenotypes that were previously reported for *pin1* null mutants, it has a strongly impaired basipetal auxin transport as shown in Figure 1 and quantitative mass spectrometric analyses indicates that at least the selected peptide regions for SRM analysis are not present in the *pin1* allele. Furthermore, immunoblots using polyclonal anti-PIN1 antibodies indicate that also the full length PIN1 protein or fractions of it are not present in the mutant.

Although it is not explicitly phrased here, we feel that also this comment refers to the interpretation of the PIN1 S4A data and the apparent weak phenotype of this transgene when introduced into the pin1 background. We hope that the additional information that we are now providing helps to clarify that the observed phenotype is exactly the phenotype that would be expected for such as transgene since PID-dependent PIN1 phosphorylation and PIN1 polarity changes required for shoot differentiation can still take place in the case of PIN1 S4A.

*6) With regard to complementation again, the reviewers had a hard time finding a clear description of* pin347 *triple mutants in the literature. You focus on this triple mutant's phototropism defect in your assays, but are there other phenotypes that could be monitored?*

The phototropism defect of the *pin3 pin4 pin7* mutant is described in the paper on the regulation of phototropism by D6PK (Willige et al., Plant Cell, 2013). We are not aware of other phenotypes of this mutant but should state at the same time that this mutant was generated in the Fankhauser laboratory (coauthors of the [37] paper) and we have agreed not to characterize this mutant further to avoid conflicts of interest with the Fankhauser lab. As already outlined in our reply to the previous comment, our main focus in the context of our manuscript was to provide evidence for a defect in auxin transport in these mutants in the presence and absence of a PIN3 wildtype or mutant transgene. This evidence is obtained by revealing the partial impairment of phototropic hypocotyl bending with the PIN3 S4A S5A variant.

7) Given the speed of the phosphorylation induced by auxin, it seems surprising that it is supposedly under control of the TIR1/AFB auxin receptors. Does this suggest that these receptors have a non-transcriptional activity in this? Then, does auxin-induced phosphorylation still occur when transcription is blocked?

Yes, the speed of changes in PIN phosphorylation is surprising, taking into account that the data at present suggest that this is mediated by TIR1/AFB. We had intuitively expected that this response would be mediated by ABP1, which is seemingly not the case. Since responses downstream of TIR1/AFBs are expected to be transcriptional responses, it would be expected that there is a delay of approximately one or two hours before auxin can efficiently interfere with signaling via TIR1/AFBs. The auxin effect through TIR1/AFBs must thus be explained by another as yet unknown mechanism. In the revised version of the manuscript, we have paid particular attention to carefully word our interpretation of this result since it cannot be explained by currently known signaling mechanisms. This definitely deserves further investigations because it may point at the existence of a novel auxin sensing signaling pathway.

We have been unable to perform the experiments with cycloheximide as suggested by the reviewer although we also consider them very important and insightful. The performance of the SRM experiments as presented in the current manuscript was already severely time-limiting experiments because the expertise in data analysis and access to the mass spectrometry equipment for these analysis is limiting. With the high qualitative standards that were applied to the experiments presented in the present manuscript, these experiments would frequently take close to a year from the moment of experimental planning, sample preparation, and sample analysis. We personally see a great potential in extending these analyses to further the understanding of auxin transport regulation per se that can give a resolution of signaling events at the protein molecular levels that is comparable to DNA sequencing. We must say that for the reasons mentioned above also many questions that could have been approached with this technology had to remain unanswered. We have therefore to excuse the omission of this critical experiment.

Should the editors and reviewers feel that the data as presented in the paper are not significant enough or too preliminary we can offer to remove these data them from the paper (Figure 10 and the corresponding supplement). At the same time, we would like to argue that this an extremely interesting, valuable and high quality data set that has already allowed us excluding some working hypotheses for ongoing further experiment, notably with regard to the regulation of this signaling event through ABP1.

8) Has it been addressed whether S4 and S5 phosphorylation are specific to activity changes of PINs or whether they also play a role in localization? Finally, do they affect the kinetics of endocytic PIN recycling?

We have examined the role of the PIN1 S4 site for PIN1 polarity, BFA-sensitivity and plasma membrane-abundance. The respective results are now presented as part of Figure 6—figure supplement 1. In brief, we have not obtained any evidence that PIN1 localization is altered in the case of the PIN1 S4A variant

In addition, we have presented related information in the original (and revised) manuscript (Figure 11) where we show by immunostaining with the anti-GFP antibody that the polarity and abundance of the wild type and S4A mutant protein are comparable. Furthermore, we have previously shown that PIN protein abundance is not altered in the *d6pk* mutants: We demonstrated that the loss of *D6PK* genes in the *d6pk012* triple mutant does not alter the abundance or localization of the PINs PIN1, PIN2, and PIN4 abundance (42). And, we had performed quantitative PIN3 immunoblots in a *d6pk* mutant series as part of our study on the role of D6PK in phototropism (37). There, we showed that there is only a slight (but at the same time reproducible) decrease in PIN3 abundance in the *d6pk012* triple and quadruple mutants. Importantly, in the *d6pk01* double mutant, which is severely impaired in phototropic hypocotyl bending and basipetal auxin transport, PIN3 levels are comparable to those of the wild type. Thus, changes in PIN3 abundance cannot explain the decrease in basipetal auxin transport in this tissue and experimental setting.

For the revised version of the manuscript, we have now discussed these different pieces of evidence more explicitly that all indicate that a reduction in PIN abundance is not sufficient to explain the reduction in PIN activity in the *d6pk* loss-of-function mutants or in lines expressing PINs that are mutated at specific PIN phosphorylation sites.

*9)*
Figure 7
*and accompanying text: if there were no difference in total abundance of PIN proteins in the* dp6k *mutants, shouldn't the sum of PIN_pS4 and PIN_S4 be the same in WT and mutants?*

The detection of a peptide by mass spectrometry is strongly dependent on its chemical composition, and even closely related peptides like PIN_pS4 and PIN_S4 may behave differentially in the mass spectrometer, as can be seen by the large differences in ion intensity values for pS and S peptides. Therefore, quantitative differences observed for a specific peptide can only be directly compared to data obtained with the same peptide in different conditions or genoyptes. It is therefore in this case not appropriate to add up quantitative data obtained with different peptides.

*10)*
Figure 8
*and accompanying text: if the S4 of PIN1 is not a target of PID/WAG kinase function, why does the S4A mutation further impair activation by PID? (compare PIN1_S1A-S3A*
^*+/-*^
*PID with PIN1_S1A-S4A*
^*+/-*^
*PID)*

This apparent discrepancy should be the consequence of the different assay systems used for the analysis of D6PK and PID function. On the one side we are using (most likely) an excess of D6PK and PID for the in vitro phosphorylation studies and in the oocyte experiments and this may affect the results of these analysis. This is why the SRM analyses as well as the analyses with the PIN1 S4A and PIN1 S4A S5A lines are particularly valuable, because they do show that in agreement with the in vitro phosphorylation experiments and the oocyte experiments, S4 and S5 do not exclusively contribute to PIN activation by D6PK (or PID). A “weakness” of the SRM analysis is that it cannot correct for the respective amount of kinase, D6PK or PID, in the sample thus the non-reduction in S4 phosphorylation in the pid sample may reflect the fact that few cells express PID in the tissue under investigation. On the other side, the strong reduction in S4 phosphorylation in the *d6pk012* mutant may be the consequence of the kinases being expressed in most of the cells in the sampled tissue. Thus, the fact that PID is expressed in comparatively few cells in the sampled tissue (or expressed at a much lower level than D6PKs) could be the cause for S4 phosphorylations not being affected in the *pid* mutant.

It is inversely also apparent from the SRM analysis of the *d6pk012* mutant where S4 and S5 phosphorylations are strongly decreased that S1 phosphorylations are not affected. Since the transgenic experiments as well as the oocyte experiments and the in vitro phosphorylation experiments all support the notion that there must be other phosphorylation sites in addition to S4 and S5 which should be S1 – S3, we must argue that S2 or S3, which we unfortunately could not analyze by SRM, are targeted by D6PKs. We have discussed this in the revised version of the paper.

*11) The reviewers are wondering whether it is true that Ler and Col behave differently from each other and that* abp1-5 *behaves differently from Ler: it seems that in all genotypes, the amount of PIN1_pS4 doubles in the presence of auxin*.

Yes, it is correct that the Ler auxin response is stronger than that observed in Col. This was also mentioned in the manuscript (original and revised). We do not know what the basis of this increased activity is.

Although auxin responses in the *abp1-5* mutant are slightly reduced when compared to the Ler wild type, the fold increase is much more reduced in the case of the *tir1/afb* mutants than in the *abp1-5* mutant. In the case of *tir1/afb* mutants, all three phosphorylations analyzed basically do not or only marginally respond to auxin treatment whereas these phosphorylations are clearly increased after auxin treatment in *abp1-5*.

[Editors' note: further clarifications were requested prior to acceptance, as described below.]

*On a side note, you might consider to remove the part describing the TIR1 pathway-dependence of PIN phosphorylation. Although, as you indicate, the data are solid, they remain unexplored with respect to the mechanism and they might distract from the main message of the paper. However, let us emphasize that it is your decision whether to keep those data in or not*.

*More importantly, an issue was arising regarding genetic background. In your assays with the* abp1 *mutant, you were using the weak* abp1-5 *allele, and it appears that you match it to a Ler wild type control. However, the originally described* abp1-5 *(Xu et al., Cell, 2010; Robert et al., Cell, 2010) was a TILLING allele obtained in the Col-0 background and cross backed to Col-0 four times. Could you please clarify the origin and nature of your allele? Finally, if the allele you used should be the described one in Col-0 background, does this change the interpretation of the results?*

We have now removed the paragraph on the analysis of the auxin-dependent effects on PIN phosphorylation in the *tir1/afb* and *abp1-5* backgrounds from the results section. Consequently, we also removed Figure 10, the respective supplements 5 and 6 as well as their legends and the reference to the mutants in the Materials and Methods section. The Discussion was also modified accordingly.

The above-mentioned changes also eliminate the issue with the *abp1-5* allele.